# Preference Goal Tuning: Post-Training as Latent Control for Frozen Policies

Guangyu Zhao [* 1]  Kewei Lian [* 2]  Haoxuan Ru [* 1]  Borong Zhang [* 1]  Haowei Lin [1]  Zhancun Mu [1]  Haobo Fu [3]
Qiang Fu [3]  Shaofei Cai [1]  Zihao Wang [1]  Yitao Liang [1]

## Abstract

Goal-conditioned policies enable decision-making models to execute diverse behaviors based on specified goals, yet their downstream performance is often highly sensitive to the choice of instructions or prompts. To bypass the limitations of discrete text prompts, we formulate post-training adaptation as a latent control problem, where the goal embedding serves as a continuous control variable to modulate the behavior of a frozen policy. We propose Preference Goal Tuning (PGT), a framework that optimizes this latent control variable to align the induced trajectory distribution with task preferences. Unlike standard fine-tuning that updates policy parameters, PGT keeps the policy frozen and updates only the latent goal from trajectory-level preferences collected through environment rollouts. This approach essentially searches for the optimal conditioning input that maximizes the likelihood of preferred behaviors while suppressing undesirable ones. We evaluate PGT on the Minecraft SkillForge benchmark across 17 tasks. With minimal data, PGT achieves average relative improvements of 72.0% and 81.6% on two foundation policies, consistently outperforming expert-crafted prompts. Crucially, by decoupling task alignment (latent goal) from physical dynamics (frozen policy), PGT surpasses full fine-tuning by 13.4% in out-of-distribution settings, demonstrating superior robustness and generalization. The project website is available at https://craftjarvis.github.io/PGT.

---

[*]Equal contribution  [1]Institute for Artificial Intelligence, Peking University  [2]School of Computing, National University of Singapore  [3]Tencent AI Lab. Correspondence to: Yitao Liang <yitaol@pku.edu.cn>.

*Proceedings of the 43$^{rd}$ International Conference on Machine Learning*, Seoul, South Korea. PMLR 306, 2026. Copyright 2026 by the author(s).

## 1. Introduction

Goal-conditioned policies pretrained on large-scale datasets have demonstrated strong capabilities in interpreting instructions and executing corresponding behaviors (Zitkovich et al., 2023; Kim et al., 2025; Cai et al., 2025b; Kareer et al., 2025). Such instructions, often referred to as "prompts", may take diverse forms, including text (Lifshitz et al., 2023; Kim et al., 2025; Wang et al., 2025b), images or videos (Wang et al., 2023a; Cai et al., 2024b), and multimodal inputs (Cai et al., 2025c; Goetting et al., 2025).

Despite their generality, the downstream performance of instruction-following policies is often highly sensitive to the choice of prompts (Lifshitz et al., 2023; Wang et al., 2024; Kim et al., 2025; Wang et al., 2023b). Identifying effective prompts typically relies on manual trial and error in a discrete text space, and prompts that appear reasonable to humans do not necessarily elicit optimal behavior.

Ideally, adaptation should effectively align the policy with specific tasks while preserving the robust, general-purpose capabilities acquired during pretraining. However, conventional approaches struggle to balance these needs. Prompt engineering respects the frozen policy but is limited by the discrete, derivative-free nature of text search, often failing to elicit optimal behaviors. Conversely, fine-tuning the policy with reinforcement learning or imitation learning can effectively shape behavior but risks catastrophic forgetting or overfitting to narrow task distributions, thereby degrading the broad generalization capabilities of the foundation model (Kirk et al., 2021; Yuan et al., 2024). This dilemma highlights the need for an adaptation mechanism that offers the optimization power of fine-tuning while maintaining the structural stability of a frozen policy.

To bridge this gap, we adopt a *latent control* perspective. We observe that a pretrained goal-conditioned policy defines a vast family of potential behaviors, indexed by its continuous goal embedding space. Rather than modifying the policy's weights—which capture the agent's pretrained dynamics priors and skill repertoire—we propose to adapt behavior solely by optimizing the *latent goal* that conditions the frozen policy. In this setting, the continuous goal embedding provides a differentiable control interface over the policy's induced trajectory distribution, allowing behavior

to be modulated more directly than by searching over discrete text prompts. Furthermore, because the optimization is restricted to a low-dimensional latent vector rather than millions of policy parameters, this approach is inherently sample-efficient and resistant to overfitting.

This formulation is closely related in spirit to soft prompt and prefix tuning, which also optimize continuous conditioning variables under a frozen backbone (Lester et al., 2021; Li & Liang, 2021; Zhou et al., 2022). However, in embodied policy adaptation, the latent goal induces an online, closed-loop trajectory distribution through environment interaction, rather than being optimized only against fixed input–output supervision. PGT exploits this structure by learning from trajectory-level preferences over positive and negative rollouts, thereby distinguishing it from supervised prompt tuning and naturally supporting a data flywheel, where the policy's own interactions provide data for iterative refinement (Wang et al., 2025a; Grannen et al., 2025; Zhu et al., 2026).

Based on this formulation, we introduce *Preference Goal Tuning* (PGT), a post-training framework that optimizes this latent control variable using preference learning. Unlike standard fine-tuning, PGT keeps the policy parameters frozen and treats the latent goal as a learnable parameter to reweight the induced trajectory distribution. Starting from an initial prompt, PGT collects a small number of trajectories, constructs pairwise preferences based on relative quality, and applies a contrastive objective to update the latent goal. This process effectively searches for the optimal conditioning input that maximizes the likelihood of preferred behaviors while suppressing undesirable ones.

We evaluate PGT on the *Minecraft SkillForge* benchmark (Johnson et al., 2016; Cai et al., 2024b) (see Appendix B.2 for details), which provides a suitable testbed for task-level adaptation: goals are semantically consistent yet executed across diverse environments. Across 17 tasks and two pretrained foundation policies, PGT consistently improves performance in both in-distribution and out-of-distribution settings, surpasses the best human-selected prompts, and demonstrates robustness to environmental variation. We further compare against parameter-efficient fine-tuning methods such as LoRA (Hu et al., 2022), BitFit (Zaken et al., 2022), and VeRA (Kopiczko et al., 2024) under the same objective, showing that PGT's out-of-distribution robustness is not merely due to fewer trainable parameters, but to adapting the frozen policy through its latent goal interface. Finally, we explore PGT as an efficient approach to continual learning, where storing a compact latent per task enables scalable adaptation without catastrophic forgetting or task interference. We also demonstrate that PGT is a general post-training mechanism applicable to foundation policies across domains, not specific to Minecraft.

## 2. Related Work

### 2.1. Goal-Conditioned Imitation Learning

Goal-conditioned imitation learning (GCIL) extends imitation learning by conditioning policies on explicit goals, enabling a single policy to execute diverse tasks given different goal inputs. Most prior GCIL methods are formulated under an end-to-end training paradigm, where both the policy and the goal representation are jointly optimized from expert demonstrations under a behavioral cloning objective.

One line of work represents goals as tokens or embeddings that are jointly modeled with states and actions in a sequence, as in large-scale instruction- or task-conditioned policies (Jang et al., 2022; Team et al., 2024; Kim et al., 2025; Black et al., 2025b;a; Cai et al., 2025b; 2026; Zhong et al., 2026; Wang et al., 2025b). Another class of approaches adopts explicit goal encoder–policy decoder architectures, where a learned goal representation conditions the policy network (Wang et al., 2023a; Lifshitz et al., 2023; Liu et al., 2025). A further line of work models goals as latent variables using variational objectives, such as conditional VAEs, jointly learning goal representations and policies in a self-supervised manner (Cai et al., 2024b; 2025c). These works demonstrate that goal-conditioned policies can support a wide range of tasks within a single model. However, a common assumption underlying these methods is the joint or end-to-end optimization of both the policy and the goal representation. This assumption can make post-training adaptation costly and may introduce interference or degradation of generalization when adapting to new tasks or environments.

In contrast, our work departs from the standard GCIL paradigm. We consider post-training adaptation of a pretrained goal-conditioned policy under a frozen policy backbone. Rather than treating the goal representation as a passive conditioning input learned during pretraining, we explicitly optimize the latent goal embedding after pretraining using trajectory-level preference feedback. This setting enables task-specific improvement while mitigating interference and preserving out-of-distribution generalization.

### 2.2. Preference Learning

Preference learning (Zhao et al., 2023b;a; Rafailov et al., 2023; Azar et al., 2024; Meng et al., 2024; Hong et al., 2024) studies how to train models from comparative or ranked feedback. It has become a central paradigm in reinforcement learning from human feedback (RLHF) (Christiano et al., 2017; Ziegler et al., 2020; Dai et al., 2024), where preference data is used to align model behavior, especially for large language models. To avoid the need for training an explicit reward model (Fürnkranz et al., 2012; Liu et al., 2022; Bai et al., 2026; Ouyang et al., 2022), prior

work has proposed preference-based learning methods that directly optimize model behavior from pairwise comparisons. Direct Preference Optimization (DPO) (Rafailov et al., 2023) and its variants optimize models directly from preference pairs under a KL-regularized objective, enabling stable and efficient alignment with a reference model. Several extensions, such as SLiC-HF (Zhao et al., 2023b;a) and IPO (Azar et al., 2024), further refine the preference objective to improve adherence to the reference policy. Beyond human-labeled preference pairs for aligning language models, preference learning has also been explored in sequential decision-making (Xia et al., 2025) and reward-based pseudo pairs (Zhang et al., 2025), where feedback is provided over a trajectory.

Most existing methods focus on the objective. In contrast, our work considers trajectory-level preference learning in goal-conditioned sequential decision-making and applies preference optimization to a different object: the latent goal embedding. By adapting preference-based objectives to optimize only the goal representation under a frozen policy backbone, our approach enables efficient post-training improvement while preserving the behavior and generalization properties of the pretrained policy.

### 2.3. Minecraft Agents and Open-World Evaluation

Minecraft is a common testbed for studying large-scale embodied agents for its open-ended environment, task diversity, and support for long-horizon decision-making. Prior work has developed agents and learning systems in Minecraft, including VPT (Baker et al., 2022), STEVE-1 (Lifshitz et al., 2023), GROOT (Cai et al., 2024b; 2025c) and ROCKET (Cai et al., 2025b; 2026; 2025a), demonstrating capabilities such as skill acquisition and generalization.

These environments have been used to evaluate a range of challenges, including long-horizon control (Jin et al., 2023; Wang et al., 2023b), precise interaction (Zhang et al., 2020; Baker et al., 2022; Cai et al., 2025b; He et al., 2025), and out-of-distribution generalization (Yang et al., 2023; Cai et al., 2023). In this work, we adopt Minecraft as an evaluation platform to assess post-training adaptation of goal-conditioned policies across diverse tasks and environments. We emphasize that our focus is not on architectural innovations for open-world exploration, but on evaluating whether optimizing latent goals can improve task performance and generalization without modifying the underlying policy.

## 3. Methodology

In this section, we introduce *Preference Goal Tuning* (PGT), a post-training framework for adapting pre-trained goal-conditioned policies through preference learning over the latent goal space. Unlike standard policy fine-tuning, PGT enforces a structural constraint: *the policy parameters are frozen*, and all adaptation is realized exclusively through the latent goal representation. An overview of the framework is shown in Figure 1.

PGT consists of two phases: (i) **preference sample generation**, where trajectories are collected and labeled according to preferences, and (ii) **behavior preference propagation**, where the latent goal is updated to favor preferred behaviors.

**Problem Formulation.** We consider a pre-trained goal-conditioned policy $\pi(a \mid s, g)$, where $g \in \mathbb{R}^d$ denotes a latent goal embedding. The policy parameters are fixed throughout post-training. This setting reflects practical constraints of foundation policies, where large-scale pre-training captures broad generalization and compositional knowledge that should not be overwritten by limited downstream supervision.

Under this constraint, behavioral adaptation must be achieved solely by adjusting the latent goal $g$, which serves as the designated control interface of the policy. Rather than learning a new policy or a reward function, our objective is to identify a latent goal that induces trajectories aligned with task-specific preferences, while remaining close to the original goal representation inferred from the initial prompt.

By restricting optimization to a low-dimensional and semantically structured latent space, PGT imposes a strong inductive bias that favors generalization over memorization, which is particularly important when only a small number of preference-labeled trajectories are available.

**Preference Sample Generation.** PGT begins with an initial prompt, which may be suboptimal, and encodes it into a latent representation $g$. Conditioned on $g$, the frozen policy $\pi(a \mid s, g)$ interacts with the environment to generate a set of trajectories, typically on the order of $\sim 10^2$.

These trajectories are then labeled according to relative preferences. When human supervision is available, annotators label each trajectory as either positive (preferred) or negative (non-preferred). Given the modest number of trajectories required, the annotation cost remains manageable.

Alternatively, in environments equipped with reward functions, such as Minecraft tasks including `collect_wood(`🟫`)`, `tool_bow(`🏹`)`, and `explore_chest(`🧰`)`, we use reward signals as a proxy for preference supervision. Specifically, trajectories with the highest cumulative rewards are treated as positive samples, while those with the lowest rewards are treated as negative samples. Importantly, reward signals are used only to induce *relative preferences* between trajectories, rather than as training targets or value estimates.

**Behavior Preference Propagation.** Given a set of positive and negative trajectories, the goal of behavior preference

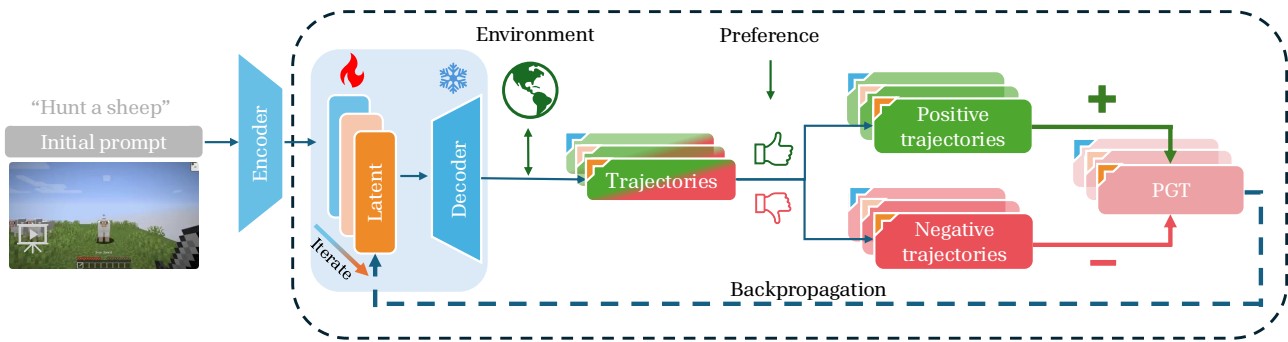

*Figure 1.* **Overview of the Preference Goal Tuning (PGT) framework.** Given a pre-trained goal-conditioned policy with frozen parameters, PGT adapts behavior solely through the latent goal. Starting from an initial prompt, a latent goal is inferred and used to collect trajectories. These trajectories are labeled with relative preferences, either by humans or by reward-based proxies. The latent goal is then updated via preference optimization, while the policy backbone remains fixed. This procedure naturally supports iterative refinement.

propagation is to update the latent goal such that preferred behaviors are encouraged and undesirable behaviors are suppressed, while keeping the policy network fixed.

*Table 1.* **Performance improvements of PGT.** All metrics are higher-is-better. We use DPO as a representative preference-learning objective. This table separates the effect of the objective, BC vs. DPO, from the effect of the adaptation target, latent-goal-only vs. full fine-tuning. BC is poorly suited to this setting, whereas DPO yields consistent gains; under DPO, latent-goal-only adaptation is comparable to or even outperforms full fine-tuning.

| Task | Latent-goal-only | | | Full Fine-Tuning | | |
|---|---|---|---|---|---|---|
| | Pretrained | BC | DPO | Pretrained | BC | DPO |
| 🪵 | 3.14 | 3.28 | **3.62** | 3.14 | 3.26 | **3.46** |
| ⬛ | 42.0 | 18.2 | **57.2** | 42.0 | 15.0 | **62.2** |
| 🪨 | 4.91 | 4.76 | **6.58** | 4.91 | 4.80 | **6.00** |
| 🟧 | 48.3 | 45.4 | **57.8** | 48.3 | 48.6 | **58.4** |

A logical starting point is to perform behavior cloning (BC) using only positive trajectories. However, as shown empirically in Table 1, this approach often fails to yield consistent performance improvements. Positive-only supervision lacks an explicit mechanism to penalize dominant but suboptimal behavior modes that are frequently encountered during rollouts, resulting in limited or unstable gains.

To solve this latent control problem without requiring a dense reward function, we leverage the probabilistic duality between control and inference. We posit that the optimal latent goal $g^*$ should induce a trajectory distribution $\pi(\tau|g)$ that minimizes the divergence from a target distribution defined by human preferences.

As derived in Appendix A, this formulation allows us to bypass explicit reward modeling. By substituting the optimal policy form into the divergence minimization objective, we obtain a direct optimization procedure over the latent space. Specifically, given pairwise preference tuples $(\tau^{(w)}, \tau^{(l)})$,

optimizing the latent goal to satisfy these preferences is mathematically equivalent to maximizing the margin between the trajectory likelihoods under the current goal $g$ and a reference goal $g_{\text{ref}}$. This yields the following tractable trajectory-wise objective:

$$\mathcal{L}(g, g_{\text{ref}}) = \mathbb{E}_{(\tau^{(w)}, \tau^{(l)}) \sim \mathcal{D}} \left[ -\log \sigma(\beta \Delta) \right],$$
$$\Delta = \sum_{i=0}^{T-1} \log \frac{\pi(a_i^{(w)}|s_i^{(w)}, g)}{\pi(a_i^{(w)}|s_i^{(w)}, g_{\text{ref}})} - \log \frac{\pi(a_i^{(l)}|s_i^{(l)}, g)}{\pi(a_i^{(l)}|s_i^{(l)}, g_{\text{ref}})}.$$
(1)

Here, $g_{\text{ref}}$ denotes a reference latent goal, and $\beta$ controls the strength of the preference signal. Crucially, optimization is performed solely over the latent goal $g$, while the policy $\pi$ remains fixed. Other preference learning objectives, such as SLiC-HF (Zhao et al., 2023b;a) and IPO (Azar et al., 2024), can be readily incorporated within the same framework. The derivation is shown in Appendix A and the result is shown in Appendix E.

Restricting optimization to the latent goal offers two key advantages. First, the latent goal serves as a semantically meaningful control interface, making it a natural locus for behavior adaptation. Second, given the limited amount of preference data, full policy fine-tuning is highly susceptible to overfitting environment-specific details. For example, in the Minecraft task collect_wood(🪵), the agent must collect logs across diverse biomes, seeds, and initial configurations. Full fine-tuning often memorizes spurious environment patterns, leading to degraded out-of-distribution generalization, whereas latent-only optimization preserves the robustness of the pre-trained policy. This robustness advantage is empirically validated in Figure 2, where PGT performs comparably to full policy fine-tuning in the in-distribution setting but consistently outperforms it under out-of-distribution evaluation.

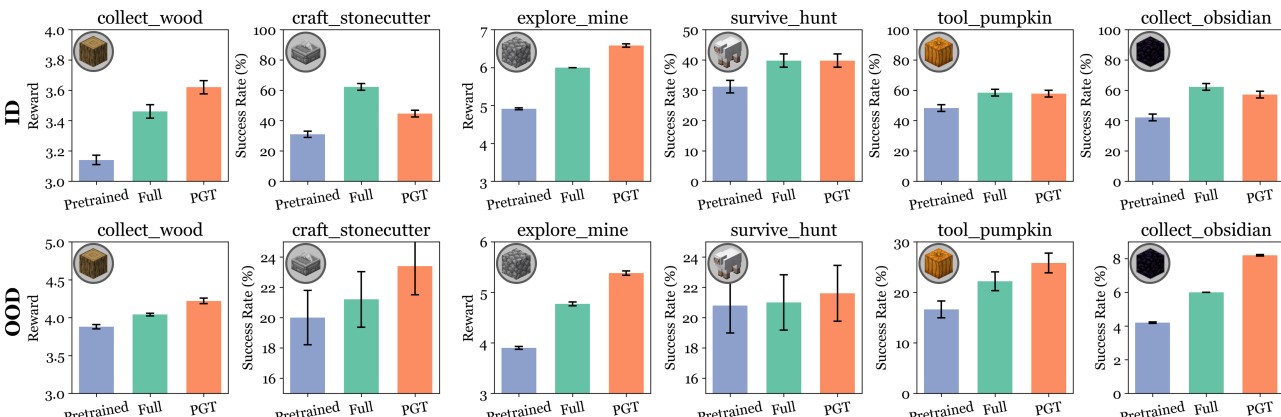

Figure 2. **Comparison between full fine-tuning and PGT.** The top row shows in-distribution (ID) performance, while the bottom row shows out-of-distribution (OOD) performance. Full fine-tuning improves in-distribution performance but often degrades generalization, whereas PGT achieves consistent gains in both settings. The definitions of both settings and the evaluation method are described in Appendix B.4 and B.5. Absolute ID and OOD scores are not directly comparable because different seeds and environment configurations can change task difficulty, e.g., resource density or distance to target objects.

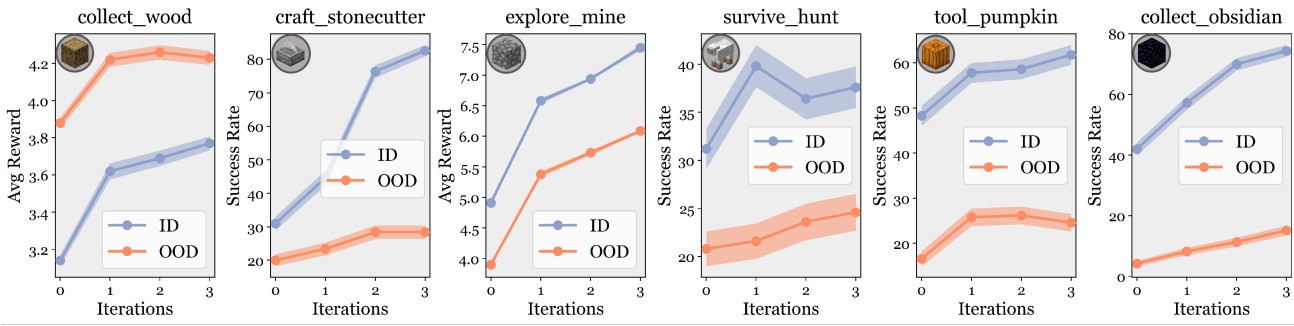

Figure 3. **Performance across iterative training rounds.** Each subplot corresponds to a task. The leftmost point represents the pre-trained policy. Blue curves denote in-distribution (ID) evaluations, while orange curves denote out-of-distribution (OOD) settings with altered initial conditions and environments. Absolute ID and OOD scores are not directly comparable. The main comparison is therefore the improvement trend within each setting, where performance consistently improves across iterations in both ID and OOD evaluations.

---

**Algorithm 1** Iterative Training PGT

1: **Input:** Policy $\pi$, Number of iterations $\mathcal{N}$, Number of samples $\mathcal{S}$, Initial latent goal $g_0$, Preference learning algorithm $\mathcal{A}$, Maximum training epoch E
2: **Output:** latent goal $g_{\mathcal{N}}$
3: **for** iteration $i \leftarrow 1$ to $\mathcal{N}$ **do**
4:     Set $g_{\text{ref}} \leftarrow g_{i-1}$
5:     Generate trajectories with $\pi$ and $g_{\text{ref}}$, choose the best and the worst $S$ ones into $\left\{\tau_s^{(w)}\right\}_{s=1}^{\mathcal{S}}$ and $\left\{\tau_s^{(l)}\right\}_{s=1}^{\mathcal{S}}$
6:     **for** epoch $e \leftarrow 1$ to E **do**
7:         Shuffle and combine $\left\{\left(\tau_s^{(w)}, \tau_s^{(l)}\right)\right\}_{s=1}^{\mathcal{S}}$
8:         Optimize $g_i$ with $\mathcal{A}$
9:     **end for**
10: **end for**
11: **return** $g_{\mathcal{N}}$

---

**Iterative Training.** The proposed framework naturally supports iterative refinement. In the first iteration, the initial prompt is encoded into a latent goal $g_0$. Both $g$ and $g_{\text{ref}}$ are initialized with $g_0$, and preference optimization yields an updated latent goal $g_1$. This updated goal is then used to recollect trajectories, from which new preference pairs are constructed.

By repeating this process, PGT performs successive improvement in the latent goal space under a fixed policy backbone. Empirically, we observe consistent performance gains for up to three iterations. Figure 3 presents detailed results across multiple tasks and both in-distribution and out-of-distribution settings. Algorithm 1 summarizes the iterative training procedure.

## 4. Experiments

We evaluate PGT on Minecraft (Fan et al., 2022; Cai et al., 2024a), a large-scale open-ended environment that poses substantial challenges for goal-conditioned policies, includ-

ing long-horizon reasoning, partial observability, and strong sensitivity to initialization and environment variations. We select tasks from the *Minecraft SkillForge* benchmark (Cai et al., 2024b), which comprises over 30 representative skills spanning 6 major categories. Additional details of the benchmark are provided in Appendix B.2.

Our experiments are designed to answer the following questions:

- Can preference-based latent goal tuning improve both in-distribution performance and out-of-distribution generalization beyond careful prompt selection?
- Does isolating task-specific adaptation to compact latent goal representations mitigate task interference and preserve generalization under sequential task acquisition?
- Can learned latent goals serve as a robust interface between high-level planners and low-level controllers in long-horizon tasks?

Across all experiments, we report results under both in-distribution (ID) and out-of-distribution (OOD) settings. OOD settings involve changes in environment seeds, initial conditions, and spatial configurations, while preserving the underlying task semantics. Absolute ID and OOD scores are not directly comparable because different seeds and environment configurations can change task difficulty, e.g., resource density or distance to target objects. We use success rate for tasks where binary completion remains discriminative, and average cumulative reward for saturated tasks such as collect_wood, where success rates above 95% make cumulative rewards a more sensitive measure of performance. Detailed descriptions of these settings are provided in Appendix B.4. Additionally, we fine-tune OpenVLA (Kim et al., 2025) on LIBERO-goal (Liu et al., 2023) and compare it with other RL- or preference-based post-training methods in Section 4.5, demonstrating the cross-domain applicability of PGT as a post-training framework.

### 4.1. Improvement Beyond Prompt Engineering

A natural baseline for adapting foundation policies is prompt engineering, where behavior is shaped by manually selecting or refining task descriptions. This experiment evaluates whether PGT provides systematic improvements beyond such initialization choices, or merely compensates for suboptimal prompts.

We discard tasks in *Minecraft SkillForge* that are either too difficult (zero success rate) or trivial (100% success rate with uninformative reward). Task selection details are described in Appendix B.3. In addition, we evaluate explore_climb( ) on GROOT as a representative subjective task; details are provided in Appendix D.

We experiment with two foundation policies, GROOT and STEVE-1, and apply DPO as a representative preference learning algorithm within PGT. Results for other preference learning objectives are reported in Appendix E. For in-distribution settings, PGT achieves average relative improvements of 72.0% on GROOT and 81.6% on STEVE-1. Under out-of-distribution conditions, the corresponding improvements are 73.8% and 36.9%, respectively. Performance gains are observed consistently across nearly all 17 evaluated tasks, with particularly notable improvements on tasks such as collect_dirt( ), craft_crafting_table( ), and tool_flint( ). Detailed per-task results are reported in Table 2.

To further examine whether the improvements from PGT depend on prompt initialization quality, we evaluate five distinct initial video prompts on the representative task collect_wood( ). For each prompt, we perform iterative preference-based training.

As shown in Figure 4, across all initial prompts—including clearly suboptimal ones—the optimized latent goals consistently surpass the best human-selected prompt, confirming that PGT's gains exceed prompt engineering. This result indicates that the gains achieved by PGT cannot be attributed to prompt engineering alone, but arise from preference-driven reshaping of the induced behavior distribution.

### 4.2. Sequential Task Adaptation

We next evaluate PGT under a sequential task acquisition setting, where tasks are learned one after another without revisiting earlier training data. In contrast to prior approaches that adapt a single shared set of policy parameters across tasks, PGT performs task-specific adaptation exclusively in the latent goal space, storing a compact latent representation for each task. As a result, task interference is avoided by construction, rather than mitigated through regularization, replay, or consolidation mechanisms. We compare PGT with multi-task learning (MTL) and several standard continual learning baselines: naive continual learning (NCL), knowledge distillation (KD) (Hinton, 2014), experience replay (ER) (Lopez-Paz & Ranzato, 2017), and elastic weight consolidation (EWC) (Kirkpatrick et al., 2017). All of them are implemented using full fine-tuning.

We first evaluate MTL on six representative tasks. As shown in Table 3, while PGT performs marginally worse than MTL in in-distribution scenarios, it achieves better aggregate performance in out-of-distribution settings, indicating improved generalization via latent goal space adaptation. We further conduct sequential continual learning in the following order: collect_obsidian( ) → tool_pumpkin( ) → craft_crafting_table( ) → explore_climb( ).

*Table 2.* **Performance of different methods on tasks in *Minecraft SkillForge*.** $\Delta$ represents the relative improvements in performance between the policy before and after post-training (represented using "+"). The tasks `collect_wood` (⬛), `collect_dirt` (⬛) and `survive_plant` (🪓) are evaluated by the collected reward (retain one decimal place), while others are expressed as success rate and the percentage sign (%) is omitted (retain two decimal places); for each task, the maximum time step for testing is 200 unless stated otherwise; the same applies to other parts of this paper. More details are in Appendix B.5.

| Task | In Distribution | | | | | | Out of Distribution | | | | | |
|---|---|---|---|---|---|---|---|---|---|---|---|---|
| | GROOT | GROOT+ | $\Delta$ | STEVE | STEVE+ | $\Delta$ | GROOT | GROOT+ | $\Delta$ | STEVE | STEVE+ | $\Delta$ |
| | 3.14 | **3.62** | 15.3% | 3.73 | **3.90** | 4.6% | 3.88 | **4.22** | 8.8% | 4.22 | **4.29** | 1.7% |
| | 27.0 | **62.8** | 132.6% | 16.3 | **36.4** | 123.3% | 15.4 | **54.6** | 254.5% | 30.4 | **48.0** | 57.9% |
| | 30.4 | **40.8** | 34.2% | 43.3 | **56.6** | 30.7% | 34.0 | **41.6** | 22.4% | 45.6 | **60.2** | 32.0% |
| | 20.2 | **20.8** | 3.0% | 4.2 | **21.8** | 419.0% | 7.8 | **9.4** | 20.5% | 41.4 | **49.0** | 18.4% |
| | 31.0 | **44.6** | 43.9% | 14.1 | **19.0** | 34.8% | 20.0 | **23.4** | 17.0% | 36.2 | **48.4** | 33.7% |
| | 5.4 | **10.4** | 92.6% | 30.9 | **40.2** | 30.1% | 4.4 | **9.6** | 118.2% | 29.6 | **41.2** | 39.2% |
| | 15.0 | **18.4** | 22.7% | 0 | 0 | - | 19.4 | **21.8** | 12.4% | 0 | 0 | - |
| | 5.4 | **14.6** | 170.4% | 4.0 | **9.6** | 140.0% | 6.0 | **18.4** | 206.7% | 2.0 | **6.4** | 220.0% |
| | 4.91 | **6.58** | 34.0% | 6.46 | **7.32** | 13.3% | 3.90 | **5.38** | 37.9% | 3.49 | **5.37** | 53.9% |
| | 15.7 | **21.2** | 35.0% | 3.4 | **4.2** | 23.5% | **38.4** | 38.2 | -0.5% | 0.5 | **0.6** | 20.0% |
| | 31.2 | **39.8** | 27.6% | **2.9** | 1.0 | -65.5% | 20.8 | **21.6** | 3.8% | **1** | 0.2 | -80.0% |
| | 31.7 | **36.6** | 15.5% | 0 | 0 | - | 83.4 | **85.6** | 2.6% | 0 | 0 | - |
| | 2.71 | **3.09** | 14.0% | 1.74 | **1.81** | 4.0% | 2.85 | **3.11** | 9.1% | 1.79 | **1.94** | 8.4% |
| | 48.3 | **57.8** | 19.7% | 1.3 | **6.2** | 376.9% | 16.6 | **25.8** | 55.4% | 7.6 | **14.0** | 84.2% |
| | 77.4 | **85.8** | 10.9% | 88.9 | **97.8** | 10.0% | 77.4 | **90.6** | 17.1% | 65.2 | **88.0** | 35.0% |
| | 1.2 | **7.4** | 516.7% | 73.6 | **76.6** | 4.1% | 1.2 | **5.8** | 383.3% | 48.0 | **52.0** | 8.3% |
| | 42.0 | **57.2** | 36.2% | 0.4 | **0.7** | 75.0% | 4.2 | **8.2** | 95.2% | 0 | 0 | - |

*Table 3.* Multi-task learning on 6 Minecraft tasks. "Ensemble" refers to fine-tuning a separate model for each task.

| Task | In Distribution (ID) | | | | Out of Distribution (OOD) | | | |
|---|---|---|---|---|---|---|---|---|
| | Pretrained | Ensemble | MTL | PGT (Ours) | Pretrained | Ensemble | MTL | PGT (Ours) |
| | 3.14 | 3.46 | **3.64** | 3.62 | 3.88 | 4.04 | **4.30** | 4.22 |
| | 31.0 | 62.2 | **66.8** | 44.6 | 20.0 | 21.2 | 18.6 | **23.4** |
| | 4.91 | 6.00 | 5.98 | **6.58** | 3.90 | 4.77 | 4.70 | **5.38** |
| | 31.2 | 39.8 | **44.2** | 39.8 | 20.8 | 21.0 | **31.4** | 21.6 |
| | 48.3 | 58.4 | **61.4** | 57.8 | 16.6 | 22.2 | 22.8 | **25.8** |
| | 42.0 | **62.2** | 53.2 | 57.2 | 4.2 | 6.0 | **10.2** | 8.2 |

The performance after learning all four tasks is reported in Table 4, with intermediate results provided in Appendix C.4, Tables 11–14.

Overall, PGT demonstrates strong robustness across tasks and environments, achieving superior out-of-distribution generalization while avoiding catastrophic forgetting. These results indicate that the advantage of PGT stems not from improved consolidation strategies, but from avoiding parameter interference altogether through latent goal isolation. While PGT does not aim to solve continual learning in the classical sense of a single shared parameterization, these results demonstrate that isolating task adaptation to latent goals effectively addresses several core challenges commonly encountered in continual learning, including catastrophic forgetting and negative transfer.

### 4.3. Long-Horizon Tasks with Planner–Controller Decomposition

We further evaluate PGT in a long-horizon setting by combining a high-level planner with a low-level controller. Specifically, we integrate the GROOT agent with the JARVIS-1 planner (Wang et al., 2024) to perform item crafting tasks from scratch in a forest environment with random initial orientations.

The planner generates high-level action sequences, while the controller executes them under the guidance of the learned goal. Agents are allowed 1000 timesteps, and we evaluate five representative items along the wood-related technology tree. Results are reported in Table 5.

Compared to the baseline, PGT consistently improves long-

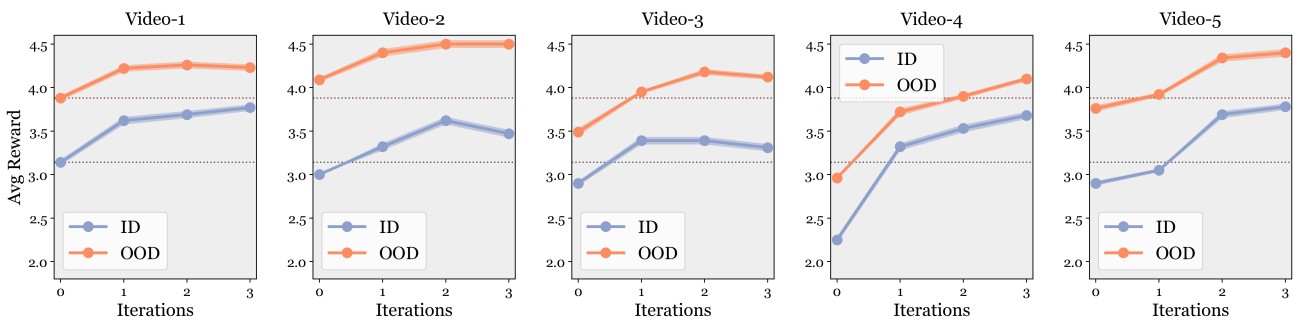

*Figure 4.* **PGT outperforms prompt engineering regardless of the initial prompt.** Each subplot corresponds to a distinct video prompt used to initialize the latent goal on `collect_wood`. Blue and orange curves denote in-distribution (ID) and out-of-distribution (OOD) performance across iterative training rounds; the dashed horizontal lines mark the *best* human-selected prompt.

*Table 4.* **Sequential continual learning, evaluated on three Minecraft tasks.** PGT is compared against four full-fine-tuning baselines (ER, EWC, KD, NCL) after sequentially learning `collect_obsidian`(⬢) → `tool_pumpkin`(🎃) → `craft_crafting_table`(📦) → `explore_climb`(🧗). The last one is used only as the final training step and is not evaluated here.

| Task | In Distribution (ID) | | | | | Out of Distribution (OOD) | | | | |
|---|---|---|---|---|---|---|---|---|---|---|
| | ER | EWC | KD | NCL | PGT (Ours) | ER | EWC | KD | NCL | PGT (Ours) |
| ⬢ | 60.2 | 64.6 | **66.8** | 61.2 | 57.2 | 6.0 | 5.4 | 5.4 | 6.8 | **8.2** |
| 🎃 | **65.4** | 60.0 | 60.8 | 61.4 | 57.8 | 25.0 | 23.8 | 20.6 | 20.4 | **25.8** |
| 📦 | 8.6 | 6.8 | 6.8 | 7.2 | **14.6** | 9.0 | 7.4 | 5.8 | 7.0 | **18.4** |

horizon task success. These results suggest that PGT-trained latent goals can serve as a behavior-level alignment interface between reactive planners and visuomotor controllers, improving robustness without modifying either component.

*Table 5.* **Success rates (%) on long-horizon tasks: crafting items from scratch.** The latent goal matches that of GROOT+ in Table 2. Results are evaluated over a total of 1,000 trials across 2 different seeds. The maximum number of steps for testing long-horizon tasks is 2000.

| Task | 🗡 | ⚔ | 🪵 | 🟫 | 📦 |
|---|---|---|---|---|---|
| Pretrained | 99.5 | 94.0 | 80.7 | 60.8 | 37.8 |
| PGT (Ours) | **100** | **100** | **89.5** | **80.7** | **64.9** |

### 4.4. Ablation Study on Parameter-Efficient Fine-Tuning

Finally, we compare PGT with other parameter-efficient fine-tuning (PEFT) methods, including LoRA (Hu et al., 2022), BitFit (Zaken et al., 2022), and VeRA (Kopiczko et al., 2024). For all methods, we use identical preference data; for PEFT baselines, we fine-tune the parameters specified by each method, while PGT optimizes only the latent goal.

As shown in Figure 5, PGT achieves competitive or superior performance under both in-distribution and out-of-distribution settings.

### 4.5. Cross-Domain Validation on Robotic Manipulation

To examine whether the effectiveness of PGT is specific to Minecraft or reflects a more general post-training principle,

we further evaluate PGT in a robotic manipulation domain. Specifically, we conduct experiments using the OpenVLA policy on the LIBERO-goal benchmark, which consists of a suite of goal-conditioned tabletop manipulation tasks with sparse success-based rewards.

LIBERO-goal differs substantially from Minecraft in state representation, action space, and task structure. While Minecraft involves language-conditioned agents operating in discrete, open-ended environments, LIBERO-goal focuses on visuomotor control with precise object interactions. As a result, this setting provides a strong test of whether preference-based latent goal tuning can generalize across domains with fundamentally different inductive biases. For each task, we freeze the OpenVLA (Kim et al., 2025) policy backbone and treat the token-level word embedding as the sole adaptation interface. Initial goal embeddings are obtained using the default task descriptions provided by the benchmark. During post-training, we collect rollouts conditioned on the current goal embedding and construct preference supervision based on task success, treating successful trajectories as preferred over failed ones. We then apply the same preference optimization procedure, optimizing only the soft prompt input while keeping the Prismatic-7B backbone (Karamcheti et al., 2024) fixed.

Figure 6 summarizes the performance of PGT in LIBERO-goal benchmark (Liu et al., 2023) compared to the OpenVLA-libero-goal (Kim et al., 2025), VLA-RL (Lu et al., 2025) and GRAPE (Zhang et al., 2025). Notably, performance improvements are observed even on tasks where

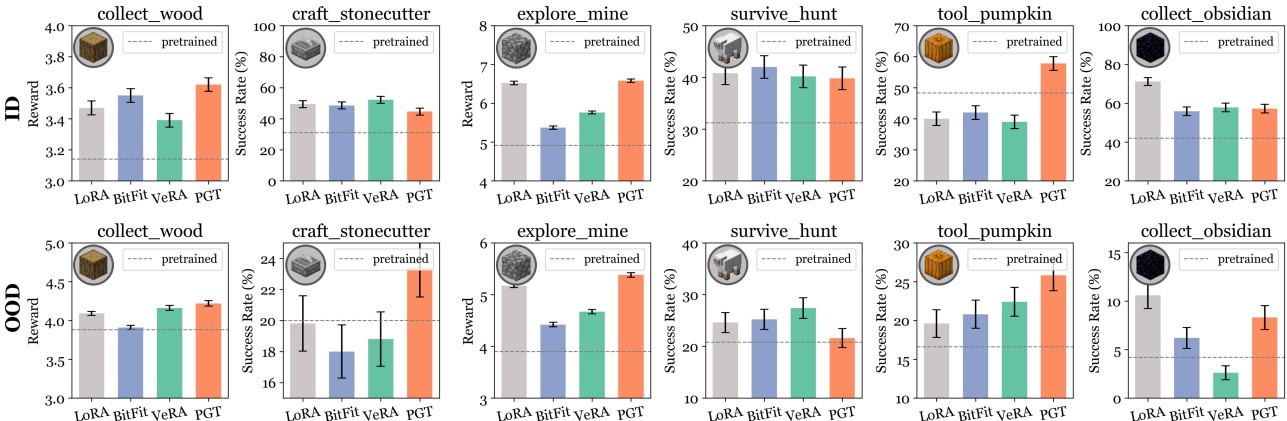

*Figure 5.* **Comparison of parameter-efficient fine-tuning (PEFT) methods.** The horizontal line denotes the performance of the pretrained GROOT model. The upper section reports results under in-distribution settings, while the lower section reports results under out-of-distribution settings. PGT demonstrates competitive and often superior performance across most settings.

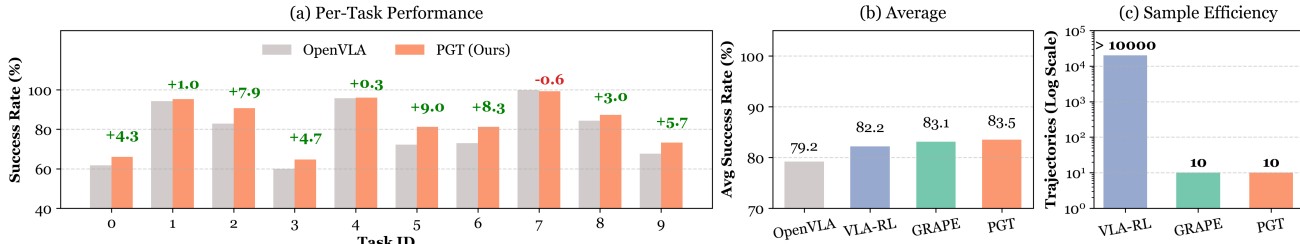

*Figure 6.* **Cross-domain validation on LIBERO-goal (Liu et al., 2023) benchmark.** PGT delivers higher performance and sample efficiency than PPO-based VLA-RL (Schulman et al., 2017; Lu et al., 2025), while matching GRAPE (Rafailov et al., 2023; Zhang et al., 2025) with a much simpler preference-ranking function.

the baseline policy already achieves strong success rates, indicating that the benefits of PGT are not limited to compensating for poor initialization. These results suggest that preference-based optimization over the latent goal space can effectively reshape induced behavior distributions without modifying the underlying policy, even in continuous-control robotic domains.

Overall, this cross-domain evaluation demonstrates that the effectiveness of PGT is not tied to the specific characteristics of Minecraft, but extends to robotic manipulation tasks with distinct state, action, and reward structures.

## 5. Conclusion and Limitations

In this work, we presented *Preference Goal Tuning* (PGT), a framework that reformulates post-training adaptation as a latent control problem. Instead of modifying the policy parameters, PGT optimizes a continuous latent goal to align the induced trajectory distribution with task preferences. By leveraging a theoretically grounded preference learning objective, PGT significantly enhances the capabilities of foundation policies with minimal data, consistently outperforming expert-crafted prompts on the *Minecraft SkillForge*

benchmark. Crucially, our experiments demonstrate that decoupling task alignment (via the latent goal) from physical dynamics (via the frozen policy) leads to superior robustness in out-of-distribution settings and effective long-horizon control when combined with high-level planners.

While PGT demonstrates remarkable effectiveness, it entails certain limitations intrinsic to its design. First, as a latent control framework, PGT operates by navigating the behavioral manifold of the frozen policy. This reliance implies that the base model must possess at least a marginal capability to execute the task; if the pretrained policy fails to sample any successful trajectories for preference construction, PGT cannot extract the necessary signal to guide optimization. Second, the optimization is task-specific: PGT learns a distinct latent control vector for each task rather than a universal generalized mapping. However, this modularity is also a strength; PGT functions as a non-destructive, plug-and-play interface that elicits optimal behaviors without altering the underlying foundation model. This ensures that the original generalization capabilities of the pretrained policy remain intact, allowing the model to handle unseen tasks seamlessly in its original zero-shot capacity.

## Acknowledgments

This work was supported by the National Science and Technology Major Project (2022ZD0114902).

## Impact Statement

This paper introduces the PGT framework for instruction-following policy post-training, which leverages a small amount of online synthesized data to efficiently enhance model performance, aiming to advance the field of machine learning. Our work carries several potential societal implications. For instance, the OOD generalization capability of the execution environment enables instruction-following robots operating in hazardous scenarios to utilize simulator data to improve their task execution in real-world settings. At present, we do not identify any ethical concerns that require special emphasis.

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

# A. Theoretical Analysis: PGT as Frozen-Family (Manifold) Latent Control

We derive the objective optimized by PGT as a form of *latent control* inside a *frozen* goal-conditioned policy. The key idea is to keep the policy parameters fixed and search over a low-dimensional conditioning variable $g$ so as to match an energy-defined target behavior implied by preferences.

## A.1. Setup: Frozen goal-conditioned policy induces trajectory distributions

Consider a controlled Markov process with state space $\mathcal{S}$, action space $\mathcal{A}$, horizon $T \in \mathbb{N}$, initial-state distribution $p_0(s)$, and dynamics $p(s' \mid s, a)$. Let $\pi_\theta(a \mid s, g)$ be a *frozen* goal-conditioned policy with parameters $\theta$ and latent conditioning $g \in \mathcal{G} \subset \mathbb{R}^d$.

We will work with *trajectory distributions* induced by $\pi_\theta(\cdot \mid \cdot, g)$. For a length-$T$ trajectory

$$\tau := (s_0, a_0, s_1, a_1, \ldots, s_{T-1}, a_{T-1}, s_T),$$

define the induced trajectory density

$$p_{\theta,g}(\tau) := p_0(s_0) \prod_{t=0}^{T-1} \pi_\theta(a_t \mid s_t, g) \, p(s_{t+1} \mid s_t, a_t). \tag{2}$$

We also fix a *reference* latent goal $g_{\text{ref}} \in \mathcal{G}$ and define

$$p_{\text{ref}}(\tau) := p_{\theta, g_{\text{ref}}}(\tau). \tag{3}$$

**Assumption 1 (Shared environment dynamics).** The conditioning $g$ affects actions only through the policy $\pi_\theta(a \mid s, g)$. In particular, $p_0(s)$ and $p(s' \mid s, a)$ do not depend on $g$. This ensures that likelihood ratios $p_{\theta,g}(\tau)/p_{\theta, g_{\text{ref}}}(\tau)$ cancel all dynamics terms.

## A.2. Frozen behavioral family ("manifold") and safety prior

The frozen policy backbone induces a family of behaviors indexed by $g$:

$$\Pi_\theta := \{\pi_\theta(\cdot \mid \cdot, g) \, : \, g \in \mathcal{G}\}. \tag{4}$$

(One may call $\Pi_\theta$ a "behavioral manifold" only after specifying a topology/geometry under which the map $g \mapsto \pi_\theta(\cdot \mid \cdot, g)$ is a smooth embedding; we will only require it as a restricted family.)

**Assumption 2 (Support/safety prior).** For all $g \in \mathcal{G}$, trajectories sampled from $p_{\theta,g}(\tau)$ remain in the support of valid, physically coherent behaviors captured by pretraining. Thus adaptation is posed as *search over $g$* rather than changing $\theta$.

## A.3. Energy-based target distribution relative to a reference prior

We encode the task not via an explicit reward but via an (unknown) energy functional $E_{\text{task}}(\tau) \in \mathbb{R}$, observed only through preferences. Following the standard maximum-entropy / control-as-inference form, define the *target* trajectory distribution

$$p^\star(\tau) := \frac{1}{Z(\beta)} \, p_{\text{ref}}(\tau) \, \exp\big(-\beta \, E_{\text{task}}(\tau)\big), \qquad \beta > 0, \tag{5}$$

where $Z(\beta)$ is the partition function. The temperature $\beta$ sets the energy scale; in preference modeling it also plays the role of a noise/inconsistency scale, so it can be treated as a hyperparameter (or estimated) without loss of generality.

## A.4. Latent control as (approximate) reweighting: realizability or projection

The family $\{p_{\theta,g}\}_{g \in \mathcal{G}}$ generally cannot represent $p^\star$ exactly. We therefore separate two cases:

**(i) Realizability (strong form).** There exists $g^\star \in \mathcal{G}$ such that $p_{\theta, g^\star}(\tau) = p^\star(\tau)$ for all $\tau$. Under realizability, Eq. (5) implies

$$-\beta E_{\text{task}}(\tau) = \log \frac{p_{\theta, g^\star}(\tau)}{p_{\text{ref}}(\tau)} + \log Z(\beta). \tag{6}$$

**(ii) Projection (weak form; what we optimize).** Without realizability, we interpret PGT as selecting

$$g^\star \in \arg\min_{g \in \mathcal{G}} D_{\mathrm{KL}}\big(p^\star(\tau) \,\|\, p_{\theta,g}(\tau)\big), \tag{7}$$

or equivalently as maximum-likelihood fitting of a preference model (next subsection). In either case, the natural *model energy* induced by $g$ is the log-likelihood ratio (up to an additive constant):

$$\widehat{E}_g(\tau) := -\frac{1}{\beta} \log \frac{p_{\theta,g}(\tau)}{p_{\mathrm{ref}}(\tau)} \quad \text{(defined up to an additive constant in } \tau\text{).} \tag{8}$$

This definition makes explicit the "latent control" interpretation: changing $g$ reweights the reference prior by a trajectory-wise likelihood ratio.

### A.5. Preferences via Bradley–Terry and the DPO-like objective over $g$

We observe only pairwise preferences $\mathcal{D} = \{(\tau_w, \tau_l)\}$, where $\tau_w$ is preferred to $\tau_l$. Under the Bradley–Terry–Luce (BTL) / logistic choice model,

$$\mathbb{P}(\tau_w \succ \tau_l) = \sigma\Big(-\beta\big(E_{\mathrm{task}}(\tau_w) - E_{\mathrm{task}}(\tau_l)\big)\Big), \qquad \sigma(u) = \frac{1}{1 + e^{-u}}. \tag{9}$$

If realizability holds, substitute Eq. (6) and note that $\log Z(\beta)$ cancels between $\tau_w, \tau_l$. More generally, in the projection view we *model* energies via $\widehat{E}_g$ from Eq. (8), leading to the model preference probability

$$\begin{aligned}
\mathbb{P}_g(\tau_w \succ \tau_l) &= \sigma\Big(-\beta(\widehat{E}_g(\tau_w) - \widehat{E}_g(\tau_l))\Big) \\
&= \sigma\left(\log \frac{p_{\theta,g}(\tau_w)}{p_{\mathrm{ref}}(\tau_w)} - \log \frac{p_{\theta,g}(\tau_l)}{p_{\mathrm{ref}}(\tau_l)}\right).
\end{aligned} \tag{10}$$

Maximizing likelihood over $\mathcal{D}$ (equivalently minimizing negative log-likelihood) yields the trajectory-wise DPO-like loss

$$\mathcal{L}(g) := -\mathbb{E}_{(\tau_w, \tau_l) \sim \mathcal{D}}\left[\log \sigma\left(\log \frac{p_{\theta,g}(\tau_w)}{p_{\mathrm{ref}}(\tau_w)} - \log \frac{p_{\theta,g}(\tau_l)}{p_{\mathrm{ref}}(\tau_l)}\right)\right]. \tag{11}$$

### A.6. Markov factorization and step-wise log-ratio form

Using Eq. (2) and Assumption 1 (shared dynamics), the trajectory likelihood ratio becomes

$$\log \frac{p_{\theta,g}(\tau)}{p_{\mathrm{ref}}(\tau)} = \sum_{t=0}^{T-1} \log \frac{\pi_\theta(a_t \mid s_t, g)}{\pi_\theta(a_t \mid s_t, g_{\mathrm{ref}})}, \tag{12}$$

since $p_0$ and $p(\cdot \mid \cdot, \cdot)$ cancel identically. Substituting Eq. (12) into Eq. (11) yields the practical step-wise objective

$$\mathcal{L}(g) = -\mathbb{E}_{(\tau_w, \tau_l) \sim \mathcal{D}}\left[\log \sigma\left(\sum_{t=0}^{T-1}\left[\log \frac{\pi_\theta(a_t^{(w)} \mid s_t^{(w)}, g)}{\pi_\theta(a_t^{(w)} \mid s_t^{(w)}, g_{\mathrm{ref}})} - \log \frac{\pi_\theta(a_t^{(l)} \mid s_t^{(l)}, g)}{\pi_\theta(a_t^{(l)} \mid s_t^{(l)}, g_{\mathrm{ref}})}\right]\right)\right]. \tag{13}$$

### A.7. Reduction to generic pairwise preference learning

To connect with standard analyses of preference learning, abstract a generic *context* $x$ and outputs $(y^+, y^-)$. Let a model $p_\varphi(y \mid x)$ and reference $p_{\mathrm{ref}}(y \mid x)$ be given, and define

$$\Delta_\varphi(x; y^+, y^-) := \log p_\varphi(y^+ \mid x) - \log p_\varphi(y^- \mid x), \tag{14}$$

$$\Delta_{\mathrm{ref}}(x; y^+, y^-) := \log p_{\mathrm{ref}}(y^+ \mid x) - \log p_{\mathrm{ref}}(y^- \mid x), \tag{15}$$

$$h_\varphi(x; y^+, y^-) := \Delta_\varphi(x; y^+, y^-) - \Delta_{\mathrm{ref}}(x; y^+, y^-) = \log \frac{p_\varphi(y^+ \mid x)\, p_{\mathrm{ref}}(y^- \mid x)}{p_\varphi(y^- \mid x)\, p_{\mathrm{ref}}(y^+ \mid x)}. \tag{16}$$

The per-pair DPO loss is

$$\ell_{\mathrm{DPO}}(h; \beta) := -\log \sigma(\beta h) = \log(1 + e^{-\beta h}), \qquad \beta > 0. \tag{17}$$

Our trajectory objective in Eq. (11) is exactly this form with the instantiation

$$\varphi \leftarrow g, \quad y \leftarrow \tau, \quad x \leftarrow \text{(initial state / prompt / context)}, \quad p_\varphi(\cdot \mid x) \leftarrow p_{\theta,g}(\cdot \mid x), \quad p_{\mathrm{ref}} \leftarrow p_{\theta,g_{\mathrm{ref}}}.$$

A.7.1. DPO $\Rightarrow$ IPO VIA A SECOND-ORDER EXPANSION WITH A UNIFORM REMAINDER BOUND

Let $f(h) := \log(1 + e^{-\beta h})$. Then

$$f'(h) = -\beta\,\sigma(-\beta h), \tag{18}$$
$$f''(h) = \beta^2\,\sigma(\beta h)\big(1 - \sigma(\beta h)\big), \tag{19}$$
$$f^{(3)}(h) = \beta^3\,\sigma(\beta h)\big(1 - \sigma(\beta h)\big)\big(1 - 2\sigma(\beta h)\big). \tag{20}$$

Since $\sigma(t)(1 - \sigma(t)) \le 1/4$ and $|1 - 2\sigma(t)| \le 1$, we have the global bound

$$\sup_{h \in \mathbb{R}} |f^{(3)}(h)| \le \frac{\beta^3}{4}. \tag{21}$$

**Theorem A.1** (Second-order Taylor approximation of DPO with a uniform cubic remainder bound). *For any $h \in \mathbb{R}$,*

$$\ell_{\mathrm{DPO}}(h; \beta) = \log 2 - \frac{\beta}{2}h + \frac{\beta^2}{8}h^2 + R_3(h), \qquad |R_3(h)| \le \frac{\beta^3}{24}|h|^3. \tag{22}$$

*Equivalently, completing the square yields*

$$\ell_{\mathrm{DPO}}(h; \beta) = \frac{\beta^2}{8}\left(h - \frac{2}{\beta}\right)^2 + \left(\log 2 - \frac{1}{2}\right) + R_3(h), \qquad |R_3(h)| \le \frac{\beta^3}{24}|h|^3. \tag{23}$$

*Proof.* Apply Taylor's theorem with Lagrange remainder at 0: $f(h) = f(0) + f'(0)h + \frac{1}{2}f''(0)h^2 + \frac{1}{6}f^{(3)}(\xi)h^3$ for some $\xi$ between 0 and $h$. Compute $f(0) = \log 2$, $f'(0) = -\beta/2$, $f''(0) = \beta^2/4$, and use (21). $\square$

The sampled IPO objective uses a squared loss

$$\ell_{\mathrm{IPO}}(h; \tau) := \tau\left(h - \frac{1}{2\tau}\right)^2, \qquad \tau > 0. \tag{24}$$

Setting $\tau = \beta/4$ aligns the minimizer of the quadratic approximation (at $h^\star = 2/\beta$) with IPO's target $h^\star = 1/(2\tau)$, and yields

$$\frac{\beta^2}{8}\left(h - \frac{2}{\beta}\right)^2 = \frac{\beta}{2}\,\ell_{\mathrm{IPO}}(h; \beta/4). \tag{25}$$

A.7.2. DPO $\Rightarrow$ HINGE RANKING TERM VIA SANDWICH BOUNDS AND A HARD-MARGIN LIMIT

Define $H(z) := \max(0, -z)$. For all $t \in \mathbb{R}$,

$$\max(0, -t) \;\le\; \log(1 + e^{-t}) \;\le\; \max(0, -t) + \log 2. \tag{26}$$

Applying (26) to $t = \beta z$ gives

$$0 \;\le\; \ell_{\mathrm{DPO}}(z; \beta) - \beta H(z) \;\le\; \log 2, \qquad \forall z \in \mathbb{R}, \tag{27}$$

and hence

$$0 \;\le\; \frac{1}{\beta}\ell_{\mathrm{DPO}}(z; \beta) - H(z) \;\le\; \frac{\log 2}{\beta}. \tag{28}$$

Thus for fixed $z$, $\lim_{\beta \to \infty} \frac{1}{\beta}\ell_{\mathrm{DPO}}(z; \beta) = H(z)$.

Since $h_\varphi = \Delta_\varphi - \Delta_{\mathrm{ref}}$, the hard-margin limit yields the hinge ranking term

$$H(h_\varphi) = \max(0, -h_\varphi) = \max\big(0,\, \Delta_{\mathrm{ref}} - \Delta_\varphi\big). \tag{29}$$

If one replaces the (possibly sample-dependent) margin $\Delta_{\mathrm{ref}}$ by a chosen constant $\delta$, this becomes

$$\ell_{\mathrm{rank}}(\varphi) = \max\big(0,\, \delta - \log p_\varphi(y^+ \mid x) + \log p_\varphi(y^- \mid x)\big), \tag{30}$$

which matches the SLiC-HF ranking/calibration term.

A.7.3. SLiC-HF REGULARIZER AS A LAGRANGIAN RELAXATION WITH AN EXACT REVERSE-KL INTERPRETATION

SLiC-HF adds a cross-entropy term on a target $y_{\text{ref}}$:

$$\ell_{\text{SLiC-HF}}(\varphi) = \max\big(0, \delta - \log p_\varphi(y^+ \mid x) + \log p_\varphi(y^- \mid x)\big) - \lambda \log p_\varphi(y_{\text{ref}} \mid x). \tag{31}$$

**Proposition A.2** (Cross-entropy regularization equals reverse KL up to a constant). *Let* $x \sim \rho(x)$ *and* $y \sim p_{\text{ref}}(y \mid x)$. *Then*

$$\mathbb{E}\big[-\log p_\varphi(y \mid x)\big] = D_{\text{KL}}\big(p_{\text{ref}}(\cdot \mid x) \,\|\, p_\varphi(\cdot \mid x)\big) + \mathbb{H}\big(p_{\text{ref}}(\cdot \mid x)\big), \tag{32}$$

*where* $\mathbb{H}(p_{\text{ref}}(\cdot \mid x))$ *is independent of* $\varphi$.

*Proof.* By definition, $D_{\text{KL}}(p_{\text{ref}}\|p_\varphi) = \mathbb{E}_{p_{\text{ref}}}[\log p_{\text{ref}} - \log p_\varphi] = \mathbb{E}_{p_{\text{ref}}}[-\log p_\varphi] - \mathbb{E}_{p_{\text{ref}}}[-\log p_{\text{ref}}]$. Rearrange and note $\mathbb{E}_{p_{\text{ref}}}[-\log p_{\text{ref}}] = \mathbb{H}(p_{\text{ref}})$. $\square$

**Exact Lagrangian form.** Consider the constrained problem

$$\min_\varphi \; \mathbb{E}\big[\ell_{\text{rank}}(\varphi)\big] \quad \text{s.t.} \quad \mathbb{E}_{(x,y)\sim\rho\, p_{\text{ref}}}\big[-\log p_\varphi(y \mid x)\big] \leq c. \tag{33}$$

Its Lagrangian relaxation is

$$\min_\varphi \; \mathbb{E}\big[\ell_{\text{rank}}(\varphi)\big] + \lambda\Big(\mathbb{E}_{\rho\, p_{\text{ref}}}[-\log p_\varphi(y \mid x)] - c\Big), \qquad \lambda \geq 0, \tag{34}$$

which matches (31) up to constants. By Proposition A.2, the regularizer equals a reverse-KL penalty $D_{\text{KL}}(p_{\text{ref}}\|p_\varphi)$ plus an additive constant, precisely formalizing "staying close to the reference."

**Implications for PGT.** Instantiating $\varphi \leftarrow g$ and $p_\varphi \leftarrow p_{\theta,g}$ shows that DPO/IPO/SLiC-HF are interchangeable preference-fitting modules for selecting $g$ inside a frozen family. They share the same core logit $h$ in Eq. (16), differing mainly in (i) the convex surrogate (logistic/quadratic/hinge) and (ii) whether reference-closeness is enforced implicitly via $\Delta_{\text{ref}}$ or explicitly via a KL-equivalent regularizer.

## B. Experiment Details

### B.1. Minecraft

Minecraft is a popular sandbox game that allows players to freely create and explore their world. Since Minecraft is an open-world environment, many recent works have designed agents and conducted explorations within Minecraft (Johnson et al., 2016). In this work, we conduct experiments on 1.16.5 version MineRL (Guss et al., 2019) and MCP-Reborn.

### B.2. Minecraft SkillForge Benchmark

Minecraft SkillForge Benchmark is a comprehensive task suite that covers various types of tasks in Minecraft. All tasks are categorized into six major groups:

- Collect task: these tasks are designed to evaluate an AI agent's capability in resource acquisition proficiency and spatial awareness.
- Craft task: these tasks are designed to shed light on an AI agent's prowess in item utilization, the intricacies of Minecraft crafting mechanics, and the nuances of various game mechanic interactions.
- Explore task: these tasks are designed to evaluate an AI agent's navigation proficiency, understanding of diverse environments, and intrinsic motivation for exploration.
- Survive task: these tasks are designed to analyze an AI agent's ability to ensure its survival, adeptness in combat scenarios, and capability to interact with the environment to meet basic needs.
- Tool task: these tasks are designed to deeply investigate an AI agent's capabilities in tool utilization, precision in tool handling, and contextual application of various tools to carry out specific tasks.
- Build task: these tasks are devised to evaluate an AI agent's aptitude in structural reasoning, spatial organization, and its capability to interact with and manipulate the environment to create specific structures or outcomes.

### B.3. Task Metrics and Selection

For most tasks, the environment logs the rewards when the corresponding objectives are achieved. We define tasks with a reward function greater than 0 as successful, and the frequency of successfully completing a task is referred to as the success rate. However, tasks like "collect_wood" "explore_mine" and "survive_plant" have a success rate of over 95% across different agents, and the specific values of the reward function are meaningful, reflecting the agents' capabilities in these tasks, so we use the detailed reward value as the metric.

We removed tasks that are so easy that agents can achieve a success rate of 100% while the specific value of the reward is either high enough (e.g. collect_grass) or not meaningful (e.g. survive_sleep). Also, to simplify the experiment, we removed the tasks for which the reward function cannot be directly obtained from the game, including subjective tasks (e.g. building tasks) and objective tasks where the environment does not log explicit rewards (e.g. craft_smelt). Moreover, mining obsidian is a high requirement for the agent's sensitivity to the objectives, and the agent needs to stay focused on the same goal over extended time steps to perform useful actions; therefore, we consider this task to be quite important and add it to the testing tasks apart from *Minecraft SkillForge*.

### B.4. Out-of-distribution Settings

We designed the out-of-distribution (OOD) setting with the goal of preventing the policy from overfitting to the environment and relying on it to dictate behavior. Thus, without altering the core meaning of the tasks, we made the following modifications to create the OOD setting:

- **Seed and agent location** We change the seed and spawn location in the Minecraft world to perform the same task, and then the initial observation will not be identical to the training set.
- **Biome** We change the biome of the agent while keeping the task solvable. For example, change biome from plains to forest of task tool_pumpkin().
- **Tool** We modified the auxiliary tools while ensuring the tasks remained solvable. For example, in the survive_hunt(), we replaced the iron_sword with diamond_axe.
- **Object location** We change the location of the object that the agent needs to interact with. For example, we changed the position of the stonecutter from being held in the hand to being placed in front of the agent.

For each task, we applied one or more of the aforementioned OOD modifications. It is important to note that the absolute performance in the OOD setting is not directly comparable to the baseline, as the tasks may become either easier or harder in the OOD environment.

### B.5. Hyperparameters

Our training hyperparameters in Minecraft settings are listed in Table 6. Each in-distribution experiment is trained and evaluated across three distinct scenarios, whereas out-of-distribution experiments are trained on the same scenarios but evaluated on a different set of three unseen scenarios. The reported test performance of GROOT and STEVE is averaged over 1,000 evaluation runs, while the results for GROOT+ and STEVE+ are averaged over 500 runs. In certain tasks, scenario generation involves manual intervention rather than relying solely on constraints natively supported by the game simulator; for example, in the survive_combat() task, mobs are explicitly spawned instead of emerging naturally from the environment.

*Table 6.* Hyperparameters for training.

| Hyperparameter | Value |
|---|---|
| Optimizer | Adam |
| Learning Rate | 1e-2 |
| $\beta$ (in DPO) | 0.6 |
| Batch Size | Full Gradient Descent |
| Type of GPUs | NVIDIA RTX 4090 |
| Training Precision | float32 |
| Number of P-N Samples (each) | 180 |

Our training hyperparameters in OpenVLA LIBERO-goal settings are listed in Table 7. Exception: We observed that OpenVLA fails excessively on Task 3 of the LIBERO-goal benchmark. To improve training stability, we increased the value of $\beta$ to 0.5.

*Table 7.* Hyperparameters for training.

| Hyperparameter | Value |
|---|---|
| Optimizer | Adam |
| Learning Rate | 1e-5 |
| $\beta$ (in DPO) | 0.3 |
| Batch Size | Full Gradient Descent |
| Type of GPUs | NVIDIA A800 |
| Training Precision | bfloat16 |
| Number of P-N Samples (each) | 10 |

# C. Experiment Results

## C.1. Behavior Cloning Results

This baseline employs behavior cloning, trained exclusively on positive samples, without the inclusion of negative data or preference learning. We present results for both tuning the latent goal only and the full parameters (Table 1).

## C.2. Full Fine-tuning Results

We compare the results of our method with full fine-tuning. The latter involves ~100M parameters, while the former only has 512 parameters, which is merely one in hundreds of thousands of the other. We found that in in-distribution settings, PGT achieves results comparable to those of full fine-tuning. However, in out-of-distribution (OOD) environments, PGT outperformed across all tasks. The results are reported in Table 8.

## C.3. Parameter-efficient Fine-tuning Results

We conduct parameter-efficient fine-tuning on LoRA (Hu et al., 2022), BitFit (Zaken et al., 2022), VeRA (Kopiczko et al., 2024), and the result is in Table 9. In fact, all of these parameter counts are significantly larger than those of PGT, and the contrast is shown in Table 10.

## C.4. Continual Learning Results

All of our continual learning baselines are based on fine-tuning the entire policy model, and the order of tasks for continual learning is as follows: collect_obsidian(⬢) → tool_pumpkin(🎃) → craft_crafting_table(🗒) → explore_climb(🧗). We conduct experiments on MTL (Table 3), NCL (Table 11), KD (Table 12), ER (Table 13), and EWC (Table 14).

*Table 8.* **Comparisons between full fine-tuning and PGT.** The "soft-prompt-like" (Lester et al., 2021; Wang et al., 2022) method can bring better improvements than the counterpart, especially on OOD settings.

| Task | In Distribution (ID) | | | Out of Distribution (OOD) | | |
|---|---|---|---|---|---|---|
| | Pretrained | Full | PGT | Pretrained | Full | PGT |
| 🧱 | 3.14 | 3.46 | **3.62** | 3.88 | 4.04 | **4.22** |
| 🧱 | 31.0 | **62.2** | 44.6 | 20.0 | 21.2 | **23.4** |
| 🪨 | 4.91 | 6.00 | **6.58** | 3.90 | 4.77 | **5.38** |
| 🐑 | 31.2 | **39.8** | **39.8** | 20.8 | 21.0 | **21.6** |
| 🎃 | 48.3 | **58.4** | 57.8 | 16.6 | 22.2 | **25.8** |
| ⬢ | 42.0 | **62.2** | 57.2 | 4.2 | 6.0 | **8.2** |

*Table 9.* Parameter-efficient fine-tuning result.

| Task | In Distribution (ID) | | | | Out of Distribution (OOD) | | | |
|---|---|---|---|---|---|---|---|---|
| | LoRA | BitFit | VeRA | PGT | LoRA | BitFit | VeRA | PGT |
| | 3.47 | 3.55 | 3.39 | **3.62** | 4.09 | 3.91 | 4.16 | **4.22** |
| | 49.4 | 48.6 | **52.2** | 44.6 | 19.8 | 18.0 | 18.8 | **23.4** |
| | 6.52 | 5.37 | 5.76 | **6.58** | 5.17 | 4.42 | 4.67 | **5.38** |
| | 39.8 | 40.8 | **42.0** | 39.8 | 24.6 | 25.2 | **27.4** | 21.6 |
| | 50.4 | 56.2 | 52.8 | **57.8** | 19.6 | 20.8 | 22.4 | **25.8** |
| | **71.2** | 55.8 | 57.8 | 57.2 | **10.6** | 6.2 | 2.6 | 8.2 |

*Table 10.* The number of trainable parameters in full fine-tuning, PGT and other baselines.

| | Full | LoRA | BitFit | VeRA | PGT |
|---|---|---|---|---|---|
| **# Parameters** | 86M | 393K | 80K | 15K | 512 |

## C.5. The Selection of $\beta$

The experiments in this paper primarily adopt DPO as an example of preference learning within the PGT framework. This algorithm involves the hyperparameter $\beta$. Figure 7 illustrates the rationale behind our choice of $\beta$. Although we selected $\beta$ based on a single task collect_wood(⬛) from a single model GROOT, we believe that the effectiveness of this framework ensures strong performance, even if the chosen hyperparameter is not optimal for other tasks or models. The results in Table 2 further confirm that $\beta = 0.6$ is indeed effective.

## D. Subjective Tasks

We also conduct experiments based on human subjective preference. The aforementioned experiments are all based on objective environmental feedback in the form of rewards, which can be seen as the "hard version" of human preferences. Additionally, we conducted validation experiments under the "soft version" as well. Considering the relatively high cost, we only conduct the experiment on explore_climb, which requires the agent to jump up the nearby hill terrain step by step, using human rankings and evaluating through TrueSkill scoring (Herbrich et al., 2006). The training set contains 40 positive and 40 negative samples, rather than 150 samples as used in objective tasks, due to the human labor cost involved in data annotation. The results are shown in Table 15. Although the confidence in the ranking of the three models' performance is not high, considering the scale of the training data, PGT can still be regarded as demonstrating a certain level of capability in subjective tasks.

## E. Other Preference Learning Algorithms

Our PGT framework consists of data filtering and preference learning. The aforementioned experiments are all based on DPO for convenience, but other preference learning algorithms like SLiC-HF (Zhao et al., 2023b;a) and IPO (Azar et al., 2024) are also possible. We experiment with them on the latent goal on several tasks and the results are listed in Table 16. It can be observed that all of them improve task performance across different environments. Different tasks are suited to different algorithms (which may also be related to hyperparameters), but performance almost consistently improves after PGT, and a latent goal with just 512-dimensional parameters is sufficient.

*Table 11.* **Task Sequential Adaptation: Continual Learning with Naive Continual Learning.** The task names in the first row represent the model trained up to the current task during sequential training (with both the pretrained model and PGT used as references); the task names in the first column represent the test results on each task. To reduce human annotation costs, we do not test the results of `explore_climb`, but use it solely as a step in the training process and reduce the number of trajectory pairs to 40. It is employed to examine the impact of later tasks on earlier ones during sequential training. The same principle applies to the subsequent tables on continual learning.

| Task | ⬢ | 🟧 | 🟫 | 🧍 | Pretrained | PGT (Ours) |
|---|---|---|---|---|---|---|
| ⬢ | 6.0 | 4.6 | 7.0 | 6.8 | 4.2 | **8.2** |
| 🟧 | | 23.6 | 24.2 | 20.4 | 16.6 | **25.8** |
| 🟫 | | | 5.2 | 7.0 | 6.0 | **18.4** |

*Table 12.* Task Sequential Adaptation: Continual Learning with Knowledge Distillation

| Task | ⬢ | 🟧 | 🟫 | 🧍 | Pretrained | PGT (Ours) |
|---|---|---|---|---|---|---|
| ⬢ | 6.0 | 5.2 | 6.6 | 5.4 | 4.2 | **8.2** |
| 🟧 | | 24.6 | 23.4 | 20.6 | 16.6 | **25.8** |
| 🟫 | | | 7.6 | 5.8 | 6.0 | **18.4** |

*Table 13.* Task Sequential Adaptation: Continual Learning with Experience Replay

| Task | ⬢ | 🟧 | 🟫 | 🧍 | Pretrained | PGT (Ours) |
|---|---|---|---|---|---|---|
| ⬢ | 6.0 | 6.6 | 5.0 | 6.0 | 4.2 | **8.2** |
| 🟧 | | 22.8 | 21.8 | 25.0 | 16.6 | **25.8** |
| 🟫 | | | 5.2 | 9.0 | 6.0 | **18.4** |

*Table 14.* Task Sequential Adaptation: Continual Learning with Elastic Weight Consolidation

| Task | ⬢ | 🟧 | 🟫 | 🧍 | Pretrained | PGT (Ours) |
|---|---|---|---|---|---|---|
| ⬢ | 6.0 | 8.2 | 5.4 | 5.4 | 4.2 | **8.2** |
| 🟧 | | 23.6 | 24.0 | 23.8 | 16.6 | **25.8** |
| 🟫 | | | 5.0 | 7.4 | 6.0 | **18.4** |

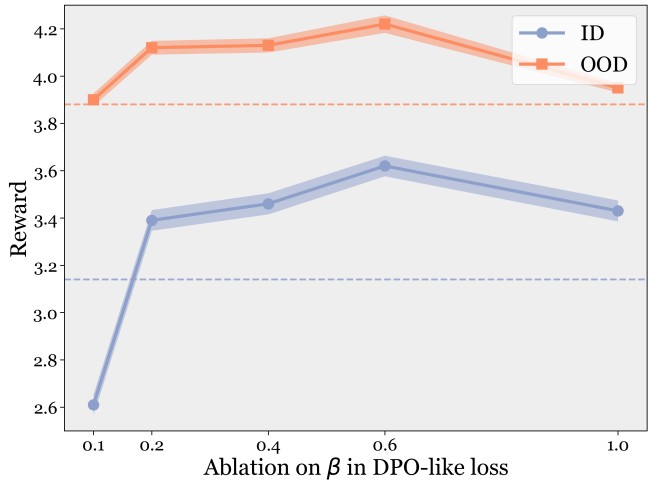

*Figure 7.* **Effect of $\beta$ on PGT.** Performance of GROOT on the tree-chopping task (`collect_wood`) under different $\beta$ values in the DPO-like loss. Blue and orange curves denote in-distribution (ID) and out-of-distribution (OOD) evaluations, respectively; the corresponding dashed horizontal lines mark the pretrained GROOT baseline in each setting. Performance peaks around $\beta = 0.6$, which we adopt as the default in our Minecraft experiments.

*Table 15.* Performance comparison on the subjective task explore_climb(🧗) using TrueSkill scores. We benchmark pretrained GROOT, DPO-fine-tuned GROOT, and GROOT with PGT (GROOT+ in Table 2) using 40 trajectory pairs for training and testing.

| 🧗 | In Distribution (ID) | | | Out of Distribution (OOD) | | |
|---|---|---|---|---|---|---|
| | Pretrained | Full | PGT (Ours) | Pretrained | Full | PGT (Ours) |
| mean($\mu$)±std($\sigma$) | 24.8±0.89 | 25.4±0.91 | **26.0±0.91** | 25.1±1.19 | 25.0±1.18 | **26.0±1.23** |

*Table 16.* PGT with other preference learning algorithms - IPO and SLiC-HF, on GROOT agent.

| Task | In Distribution (ID) | | | | Out of Distribution (OOD) | | | |
|---|---|---|---|---|---|---|---|---|
| | Pretrained | DPO | IPO | SLiC-HF | Pretrained | DPO | IPO | SLiC-HF |
| 🪵 | 3.14 | **3.62** | 3.37 | 3.24 | 3.88 | **4.22** | 3.99 | 4.00 |
| 🪨 | 31.0 | **44.6** | 42.0 | 37.0 | 20.0 | 23.4 | 23.0 | **25.6** |
| 🪨 | 4.91 | **6.58** | 5.44 | 6.34 | 3.90 | **5.38** | 4.70 | 5.29 |
| 🐑 | 31.2 | 39.8 | 40.6 | **43.0** | 20.8 | 21.6 | **32.8** | 24.4 |
| 🗄 | 48.3 | 57.8 | **62.2** | 60.6 | 16.6 | 25.8 | **30.6** | 27.6 |
| ⬛ | 42.0 | **57.2** | 50.4 | 34.4 | 4.2 | **8.2** | 4.8 | 3.4 |

