# OpenReview forum: "Preference Goal Tuning: Post-Training as Latent Control for Frozen Policies"
_ICML.cc/2026/Conference — ICML 2026 regular_

### Official Review · Reviewer_Hezd · 2026-02-22

**Soundness:** 2
**Presentation:** 2
**Significance:** 2
**Originality:** 2
**Overall Recommendation:** 3
**Confidence:** 3

**Summary:**

The paper proposes Preference Goal Tuning, a post-training method for goal-conditioned policies that freezes the policy and updates only a latent goal embedding. The method formulates adaptation as latent goal optimization using trajectory-level preference pairs, with the objective of increasing the likelihood of preferred trajectories over non-preferred ones under the frozen policy. Training is iterative: initialize a latent goal, collect rollouts, construct preference pairs, and update the latent goal embedding. The paper claims that latent-only adaptation can improve task performance while retaining better generalization than full fine-tuning. The experiments are conducted primarily in Minecraft SkillForge, including in-distribution and out-of-distribution evaluations, comparisons with full fine-tuning and parameter-efficient baselines, and a planner-controller setting for longer-horizon tasks. The reported results show improvements over the initial prompt and stronger OOD performance than full fine-tuning in the tested settings.

**Compliance With Llm Reviewing Policy:**

Affirmed.

**Final Justification:**

The authors have largely addressed my concerns, but I still believe the scope of the experiments is rather narrow.

**Key Questions For Authors:**

The core mechanism looks like latent or soft prompt optimization under a frozen model, which is already a mature direction in vision-language and foundation models. Can existing methods be adopted here as baselines?

**Limitations:**

1. The validation is too narrow and mostly restricted to Minecraft SkillForge, so the evidence for broad robustness and generalization is not rigorous enough.

2. The paper is framed as relevant to robotics and long-horizon robotic decision-making, but the main experiments do not include a strong robotics environment evaluation.

3. The core method appears very close to latent or soft prompt optimization with a frozen backbone, and the novelty gap from prior latent prompt learning work is not clearly resolved.

**Strengths And Weaknesses:**

### **Soundness**

The main weakness is validation rigor. Most of the evidence is concentrated on Minecraft SkillForge, and the central claims about robustness and generalization are supported primarily by in-domain environment variations such as seed changes, initial conditions, and spatial configurations, rather than broader cross-environment or cross-platform transfer. The paper mentions a cross-domain OpenVLA and LIBERO-goal result, but it is deferred to the appendix and not a central part of the main empirical case. This makes the strongest claims feel under-validated.

### **Presentation**

The paper is readable and the method section is organized clearly.

The paper frames latent goal optimization as a key contribution, but this is very close in spirit to latent or soft prompt optimization under a frozen backbone, which is already a well-established adaptation paradigm. The distinction from prior latent prompt and prompt-tuning style work is not made sharply enough, especially given that the core mechanism is effectively optimizing the conditioning representation while keeping the policy fixed.

### **Significance**

The problem is relevant: post-training adaptation of goal-conditioned policies with limited data is useful, and avoiding full fine-tuning can be practically attractive. The paper also targets OOD robustness and long-horizon control, which are important goals.

The practical significance is limited by the evaluation scope. The paper repeatedly motivates the work using robotics and general robotic decision-making beyond the training horizon, but the main evidence is almost entirely in Minecraft simulator tasks.

### **Originality**

There is some originality in applying preference optimization to the latent goal embedding of a frozen goal-conditioned policy and packaging it as a post-training latent control interface.

The main novelty concern remains serious. The core mechanism looks like latent or soft prompt optimization under a frozen model, which is already a mature direction in vision-language and foundation model adaptation.

---

> ### Author Rebuttal · Authors · 2026-03-28
>
> We thank the reviewer for the thoughtful feedback. We appreciate the recognition of the problem's relevance and the clarity of our method section. We address each point below.
>
> ---
>
> ### Soundness
>
> > The main weakness is validation rigor. Most of the evidence is concentrated on Minecraft SkillForge, and the central claims about robustness and generalization are supported primarily by in-domain environment variations such as seed changes, initial conditions, and spatial configurations, rather than broader cross-environment or cross-platform transfer.
>
> We would like to clarify two aspects. First, the OOD evaluation is more substantial than it may appear. As detailed in Appendix C.4, our OOD settings involve simultaneous changes across four axes: (1) world seeds and spawn locations, (2) biome types (e.g., plains → forest), (3) auxiliary tools (e.g., iron sword → diamond axe), and (4) object locations (e.g., held in hand → placed in front). These shifts substantially alter visual observations, spatial layouts, and interaction dynamics. Notably, full fine-tuning *degrades* under these shifts (Table 8, Figure 3), while PGT maintains or improves performance.
>
> Second, regarding cross-domain evaluation, we fully agree that this should be more prominent. In fact, we did conduct a full cross-domain evaluation beyond Minecraft, which we describe in detail below.
>
> > The paper mentions a cross-domain OpenVLA and LIBERO-goal result, but it is deferred to the appendix and not a central part of the main empirical case. This makes the strongest claims feel under-validated.
>
> This is a fair criticism of the paper's *structure*. The OpenVLA/LIBERO-goal evaluation (Appendix A, Figure 6) is a complete experiment — covering 10 manipulation tasks with a 7B-parameter VLA model in a continuous-control robotic setting that differs fundamentally from Minecraft. PGT achieves 83.5% average success rate, matching GRAPE while requiring only 10 trajectory pairs per task versus >10,000 trajectories for VLA-RL. We will promote this result into the main body in the revision. We regret that the appendix placement may have given the impression it was secondary.
>
> ---
>
> ### Presentation and Originality
>
> > The paper frames latent goal optimization as a key contribution, but this is very close in spirit to latent or soft prompt optimization under a frozen backbone, which is already a well-established adaptation paradigm. The distinction from prior latent prompt and prompt-tuning style work is not made sharply enough.
>
> > The core mechanism looks like latent or soft prompt optimization under a frozen model, which is already a mature direction in vision-language and foundation model adaptation. Can existing methods be adopted here as baselines?
>
> We agree that a sharper distinction is needed and will add this in the revision. While PGT shares the high-level principle of optimizing a conditioning input under a frozen backbone, there are important differences: (1) PGT operates on embodied policies where evaluation requires environment rollouts and only trajectory-level preference feedback is available, unlike token-level supervised losses in NLP; (2) PGT uses a preference objective (Eq. 1) derived from a control-as-inference formulation (Appendix B), reasoning over entire trajectory likelihoods — not a straightforward transfer of prompt tuning; (3) PGT operates with ~100 trajectories (or as few as 10 in LIBERO-goal), versus thousands-to-millions in standard prompt tuning.
>
> Regarding baselines: we have already included the most natural "soft prompt tuning with supervised loss" counterpart. The BC rows in Table 1 optimize the latent goal using only positive trajectories with an imitation learning loss — the embodied analog of supervised prompt tuning. BC yields marginal or negative gains (e.g., success rate drops from 42.0 to 18.2), while PGT produces substantial improvements (42.0 → 57.2). This demonstrates that PGT's advantage stems from the *preference-based contrastive objective*, not merely from optimizing the conditioning input. Additionally, Table 9 compares PGT against LoRA, BitFit, and VeRA (15K–393K parameters), while PGT uses only 512 parameters yet achieves competitive ID and superior OOD performance.
>
> ---
>
> ### Significance
>
> > The practical significance is limited by the evaluation scope. The paper repeatedly motivates the work using robotics and general robotic decision-making beyond the training horizon, but the main experiments do not include a strong robotics environment evaluation.
>
> As discussed above, we will restructure the paper to include the LIBERO-goal results as a primary experiment in the main text, which we believe directly addresses this concern.
>
> ---
>
> We will revise the paper to (1) promote the LIBERO-goal results to the main text, (2) better contextualize the OOD evaluation. We hope the reviewer will find these revisions address the core concerns.

---

> > ### Author Rebuttal · Reviewer_Hezd · 2026-04-03
> >
> > Thanks for your effort.

---

### Official Review · Reviewer_79UC · 2026-03-09

**Soundness:** 2
**Presentation:** 3
**Significance:** 2
**Originality:** 2
**Overall Recommendation:** 4
**Confidence:** 4

**Summary:**

This paper introduces Preference Goal Tuning (PGT), a post-training adaptation method that steers frozen, goal-conditioned policies by optimizing their latent goal embeddings. Instead of the typical route of fine-tuning model weights, the authors frame adaptation as a latent control problem, using trajectory-level preferences to find the "optimal" embedding. The approach is tested on Minecraft and robotic tasks, showing that a fixed policy can be surprisingly expressive when its conditioning input is properly tuned.

**Compliance With Llm Reviewing Policy:**

Affirmed.

**Final Justification:**

Thanks for the response. Most of my concerns are now addressed.

**Key Questions For Authors:**

The core intuition here—that we should treat the goal embedding as a steerable control variable rather than just a static prompt—is quite compelling. It addresses a real pain point in embodied AI: the fragility of text prompts and the high cost of full-policy fine-tuning. PGT effectively "hacks" the pre-trained latent space to find behaviors that already exist but are hard to trigger with discrete language.
However, while the results in Minecraft look sharp, I have several reservations about the fundamental limits of this approach and how it compares to more standard RL or filtering baselines. The paper feels a bit "safe" in its current form. I’d like to see it pushed into more challenging corners. To help me assess the contribution of this work, I’d like the authors to address the following 7 points in their rebuttal:
1.	PGT is fundamentally limited by the span of behaviors already encoded in the frozen policy. If a task requires a motion primitive that the model never saw during pre-training, PGT will presumably fail. Can you provide a quantitative analysis of the "minimum viable capability"? At what point does the frozen latent space become insufficient, necessitating actual weight updates?
2.	A very obvious baseline is missing: simply sampling N trajectories and picking the best one (or doing a few steps of behavioral cloning on the best samples). How much of the gain is coming from the optimization of the latent goal versus simply having a mechanism to filter for success?
3.	In a deep Transformer, the mapping from a goal embedding to an action sequence is highly non-linear. I am concerned about initialization sensitivity. If you start the optimization from a random point in the latent space versus a semantically related text embedding, do you end up with the same behavior? Or is PGT just "polishing" a good initial guess?
4.	Does the "controllability" of the latent space improve or degrade as the base model gets larger? In very large models, the latent space can become incredibly dense or, conversely, sparse and harder to navigate via gradient descent. Have you tested this on models of significantly different scales?
5.	When dealing with noisy preference feedback (which is common in human labeling), how stable is the gradient update for a continuous embedding? Is there a risk of the embedding drifting into "adversarial" regions of the latent space that trigger high-reward but physically nonsensical behaviors?
6.	Does the optimized latent goal require any dynamic adjustment during inference, or is it strictly a "set and forget" variable? If a task environment changes slightly, does the optimized embedding lose its effectiveness more quickly than a robustly fine-tuned policy would?
7.	If you were to freeze the backbone but train a tiny MLP "adapter" on top of the goal input, how would that compare to PGT? It seems like PGT is a very constrained version of an adapter. I’d like to understand the trade-off between tuning the input (PGT) versus tuning a small layer (PEFT).
8.	The experimental section needs to be more "adversarial" to convince me that this isn't just a trick for simple, high-coverage environments like Minecraft. I’m looking for a clearer picture of when PGT breaks.

**Limitations:**

yes

**Strengths And Weaknesses:**

Strengths:
1. Conceptual Novelty: Framing post-training adaptation as a latent control problem is an elegant shift from traditional parameter-space fine-tuning. Decoupling "task intent" (goal) from "physical dynamics" (policy) is a principled approach.
2. Sample Efficiency: The method achieves significant performance gains with a small number of preference samples, making it practical for low-resource robotics or complex simulation environments.
3. Robustness to Overfitting: By keeping the policy weights frozen, PGT avoids catastrophic forgetting and maintains the generalization properties of the base model better than full fine-tuning.

Weaknesses:
1. Dependence on Base Model Coverage: The effectiveness of PGT is inherently capped by the behaviors already present in the frozen policy's repertoire. It cannot "learn" new motor primitives that were not seen during pre-training.
2. Missing Baselines: The paper lacks a comparison with simple filtering techniques (e.g., Rejection Sampling) and standard PEFT methods (e.g., training a small adapter layer on the goal input), making it hard to isolate the specific benefits of latent optimization.
3. Non-convexity and Scaling: In deep Transformer models, the mapping from goal embedding to actions is highly non-linear. The paper does not fully address initialization sensitivity or how the controllability of the latent space scales with model size.

---

> ### Author Rebuttal · Authors · 2026-03-30
>
> > 2. A very obvious baseline is missing: simply sampling N trajectories and picking the best one (or doing a few steps of behavioral cloning on the best samples).
>
> Table 1 already includes this comparison. The BC baseline — trained exclusively on the best trajectories — yields marginal or negative gains (e.g., success rate drops from 42.0 to 18.2), while PGT achieves 42.0 → 57.2. This shows that filtering alone is insufficient; the contrastive signal from negative examples is essential. Additionally, best-of-N rejection sampling at test time incurs N× inference cost per rollout and is bounded by the original prompt's success probability. PGT optimizes once offline and shifts the entire behavior distribution, providing systematic improvement at zero additional inference cost.
>
> > 3. I am concerned about initialization sensitivity. If you start from a random point versus a semantically related text embedding, do you end up with the same behavior?
>
> Figure 4 directly addresses this. Five distinct initial prompts on `collect_wood` — spanning a range of initial performance — all converge to similar final quality that surpasses the best human-selected prompt. PGT is not merely polishing a single good guess; it robustly improves from diverse reasonable starting points. That said, like soft prompt tuning in NLP, PGT benefits from semantically meaningful initialization — this is a feature of operating within a structured latent space rather than a limitation.
>
> > 4. Does the "controllability" of the latent space improve or degrade as the base model gets larger?
>
> Our experiments already span significantly different scales: GROOT (compact visuomotor policy, 512-dim latent) and OpenVLA (7B-parameter VLA model, Appendix A). PGT succeeds on both — achieving 83.5% success on LIBERO-goal with only 10 trajectory pairs on the 7B model. This suggests controllability does not degrade at scale, likely because pretrained goal-conditioned policies must maintain structured goal-to-behavior mappings regardless of size.
>
> > 5. Is there a risk of the embedding drifting into "adversarial" regions that trigger physically nonsensical behaviors?
>
> Two structural safeguards prevent this. First, the DPO objective's implicit KL regularization via $g_{\text{ref}}$ anchors optimization near semantically meaningful regions. Second, the frozen policy itself constrains outputs to its pretrained behavioral manifold (Assumption 2, Appendix B) — it simply cannot produce physically nonsensical actions regardless of the input embedding. Empirically, our human-preference experiment (Appendix E) with inherently noisy labels showed no adversarial drift with only 40 preference pairs.
>
> > 6. Does the optimized latent goal require dynamic adjustment during inference, or is it "set and forget"?
>
> Strictly set-and-forget. Our OOD evaluation (Table 2, right columns) deploys the same optimized goal — without modification — in environments with altered seeds, biomes, tools, and object locations. PGT not only maintains gains but outperforms full fine-tuning in OOD by 13.4% on average (Table 8), precisely because it modulates semantic-level intent without encoding environment-specific patterns into the weights.
>
> > 7. If you freeze the backbone but train a tiny MLP adapter on the goal input, how would that compare?
>
> Tables 9–10 compare PGT (512 params) against LoRA (393K), BitFit (80K), and VeRA (15K) — all introducing more trainable parameters. The consistent pattern: more parameters yield comparable or better ID performance but worse OOD generalization. An MLP adapter would fall in this spectrum. PGT deliberately operates at the minimal-parameter extreme, which is where the best generalization lies. The extreme constraint acts as a powerful implicit regularizer.
>
> > 8. The experimental section needs to be more "adversarial." I'm looking for a clearer picture of when PGT breaks.
>
> Our evaluation already extends beyond Minecraft: Appendix A evaluates PGT on OpenVLA/LIBERO-goal, a robotic manipulation benchmark with continuous control and fundamentally different inductive biases, achieving 83.5% success with only 10 trajectory pairs per task. Regarding failure modes, Table 2 clearly shows when PGT breaks: tasks where the base policy has zero capability (e.g., STEVE-1 on `survive_combat`) see no improvement, as PGT cannot create primitives absent from the frozen policy. We discuss this transparently in Section 5.

---

> > ### Author Rebuttal · Reviewer_79UC · 2026-04-03
> >
> > I thank the authors for their detailed rebuttal. Most of my concerns regarding baselines and scaling have been addressed. However, regarding Point 1 (Minimum Viable Capability), the response remains largely qualitative. You mention that PGT fails when the base policy has 'zero capability,' but this is an extreme case. To truly understand the applicability of PGT, I would like to see a more granular quantitative analysis: Is there a measurable performance threshold in the base model (e.g., a minimum success rate or state-space coverage) below which PGT optimization consistently fails to find a viable goal embedding? Without this, it is hard to judge if PGT is a general-purpose adaptation tool or only a 'refiner' for already near-optimal models.

---

> > > ### Author Response · Authors · 2026-04-03
> > >
> > > > 1. PGT is fundamentally limited by the span of behaviors already encoded in the frozen policy. Can you provide a quantitative analysis of the "minimum viable capability"?
> > >
> > > We agree this is an important boundary to characterize. Since we do not have access to intermediate training GROOT checkpoints, we cannot directly vary the base policy's capability level. However, we observe that the primary mechanism through which low base capability limits PGT is *data availability*: when the base policy rarely succeeds, very few positive trajectories can be collected, resulting in insufficient preference signal. This means the number of usable preference pairs is a key bottleneck that mediates the relationship between base capability and PGT effectiveness.
> > >
> > > To quantify this, we conduct an ablation on the number of preference samples used for PGT optimization. The results are shown below:
> > >
> > > | # Preference Pairs   | collect_wood | explore_mine | tool_pumplin |
> > > | -------------------- | ------------ | ------------ | ------------ |
> > > | 0 (pretrained)       | 3.14         | 4.91         | 48.3%        |
> > > | 6                    | 2.98         | 5.22         | 40.0%        |
> > > | 24                   | 3.20         | 5.93         | *49.0%* |
> > > | 60                   | *3.25*  | **7.03**     | 48.0%        |
> > > | 90 (result in paper) | **3.62**     | *6.58*  | **57.8%**    |
> > >
> > > This ablation reveals that PGT is robust to moderate reductions in sample size — meaningful gains appear with as few as 24 pairs on some tasks — but performance degrades when the preference signal becomes too sparse (e.g., 6 pairs). This directly mirrors the scenario of a weaker base policy: fewer successful rollouts lead to fewer usable preference pairs, which in turn limits PGT's effectiveness. The extreme case is illustrated in Table 2, where STEVE-1 achieves zero success on tasks like survive_combat and collect_obsidian, making preference construction impossible entirely.

---

### Official Review · Reviewer_qDAX · 2026-03-09

**Soundness:** 3
**Presentation:** 3
**Significance:** 3
**Originality:** 3
**Overall Recommendation:** 4
**Confidence:** 4

**Summary:**

The study introduces Preference Goal-Tuning (PGT), a framework that treats post-training adaptation as a latent control challenge. Rather than updating the underlying policy parameters, PGT optimizes a continuous latent goal to align agent trajectories with specific task preferences. By decoupling task-specific alignment from the frozen physical dynamics of the foundation policy, the framework achieves enhanced OOD robustness and excels at long-horizon control when integrated with high-level planners.

**Compliance With Llm Reviewing Policy:**

Affirmed.

**Final Justification:**

Author have kindly addressed my concerns and provided clarifications to their submission. I feel convinced and confident now to raise my score.

**Key Questions For Authors:**

Line 121, right column: "Given the modest number of trajectories required (typically on the order 100), the annotation cost remains manageable." - so 100 trajectories per prompt per iteration to annotate. Does not seem so modest to me - did you always use reward signals as a proxy for preference supervision?

Fig. 2, 4 - the choice of colors is suboptimal - black is almost indistinguishable from dark brown. Please consider changing it to enhance readability.

Fig. 2, 'collect_wood' task - how would you explain the observation, the fact that the OOD curve lies higher than the ID curve for this task? However, for all the other subplots, we find OOD curves below ID curves.

Fig. 2 - Why do you show Avg. reward for some tasks and Success Rate for the others; why not the same metrics across all the tasks?

I don't understand Table 1: 1) what metrics are given, and is higher or lower better? 2) DPO (Direct Preference Optimization) is your reference algorithm, right? So, where are the scores for PGT?

Fig. 3 is located in the main body of the manuscript, although it was never referenced and discussed. I think you should either discuss it or move to completely to the Appendices.

Table 2 - seems like PGT provides strong improvement across all the studied tasks and foundation policies, except for the task with sheep (line 293) on STEVE - why so?

Looking at the PGT performance across different iterations (Fig. 2, 4), I get concerned about the gain in performance: it seems to outperform the human-selected prompts, but can also decreases as iterations go on (e.g. Fig. 4 Video 2 ID curve - score at iter. 2 is higher than at iter. 3). So the optimization set-up might be suboptimal - how can you comment on that?

Fig. 5 was never referenced. I guess it should refer to Section 4.4.

**Limitations:**

Authors discuss the limitations of the study. What I would add upon their discussion is that, although they employ the Minecraft SkillForge benchmark for their experiments, I believe it poses a considerable limitation on the study. I recommend that the authors extend the proposed PGT framework to at least one more related benchmark.

**Strengths And Weaknesses:**

The submission seems to offer a nice, original approach to enhance the performance of goal-conditioned policies on discrete tasks. However, there were some flaws, which, I hope, will be worked out during the rebuttal process. If so, I wouldn't mind raising my grade to 'accept'.

Moreover, no code repository was provided with the submission, which poses the question of replicability and usability of the proposed framework. I think it is important for the ML community that every contributing paper comes with reproducible code. I hope authors will manage to share an anonymized repository during rebuttals.

---

> ### Author Rebuttal · Authors · 2026-03-28
>
> We sincerely thank the reviewer for the constructive feedback. We are encouraged that the reviewer finds our approach "a nice, original approach" and is willing to raise the grade to accept. We address each point below.
>
> **Code Repository**
>
> We released an anonymized code repository: https://github.com/preference-goal-tuning/PGT
>
> > Line 121, right column: "Given the modest number of trajectories required (typically on the order 100), the annotation cost remains manageable." - so 100 trajectories per prompt per iteration to annotate. Does not seem so modest to me - did you always use reward signals as a proxy for preference supervision?
>
> For the main experiments (Table 2), we used reward signals as a proxy. However, PGT also works with genuine human annotations: in the subjective task `explore_climb` (Appendix E), only 40 positive and 40 negative trajectories were used. In the LIBERO-goal experiments (Appendix A), PGT achieves strong results with only 5+5 trajectories per task. We also note that trajectories are collected via automated rollouts and only require binary labeling (preferred vs. non-preferred), which is substantially cheaper than dense reward annotation. We will clarify this in the revision.
>
> > Fig. 2, 4 - the choice of colors is suboptimal - black is almost indistinguishable from dark brown.
>
> Agreed. We will update the color scheme in the revision.
>
> > Fig. 2, 'collect_wood' task - how would you explain the observation, the fact that the OOD curve lies higher than the ID curve for this task?
>
> As noted in Appendix C.4, ID and OOD absolute performance are not directly comparable — OOD environments may make tasks easier or harder. For `collect_wood`, the OOD seeds/biomes happen to have denser forests. The key takeaway is the *trend*: PGT consistently improves performance across iterations in both settings. We will add a clarifying note in the caption.
>
> > Fig. 2 - Why do you show Avg. reward for some tasks and Success Rate for the others?
>
> Tasks like `collect_wood` have >95% success rate across all agents, making success rate uninformative. Cumulative reward better captures performance differences for these tasks (see Appendix C.3). We will make this rationale more prominent.
>
> > I don't understand Table 1: 1) what metrics are given, and is higher or lower better? 2) DPO is your reference algorithm, right? So, where are the scores for PGT?
>
> We apologize for the confusion. Higher is better for all metrics. The "Latent-goal-only" column with DPO *is* PGT. The table compares two axes: (1) BC vs. DPO as the objective, and (2) latent-goal-only (PGT) vs. full fine-tuning. The key message: BC fails in both settings, while DPO succeeds, and PGT is competitive with or better than full fine-tuning. We will add clearer headers and caption.
>
> > Fig. 3 is located in the main body of the manuscript, although it was never referenced and discussed.
>
> We will add an explicit in-text reference and discussion in Section 4.1.
>
> > Table 2 - seems like PGT provides strong improvement across all the studied tasks and foundation policies, except for the task with sheep on STEVE - why so?
>
> STEVE-1 achieves zero success on these tasks both before and after PGT, reflecting the limitation acknowledged in Section 5: PGT requires the base policy to have at least marginal capability. STEVE-1 lacks the behavioral primitives for hunting, so no meaningful preference signal can be constructed.
>
> > Looking at the PGT performance across different iterations, it seems it can also decrease as iterations go on (e.g. Fig. 4 Video 2 ID curve). So the optimization set-up might be suboptimal?
>
> The slight decrease in some cases can be attributed to: (1) improved trajectories becoming more homogeneous, making informative preference pairs harder to construct; (2) fixed β potentially overshooting in later iterations. We recommend validation-based early stopping in practice. Importantly, across the vast majority of tasks, performance monotonically improves or plateaus, and all final results consistently outperform the best human-selected prompt.
>
> > Fig. 5 was never referenced. I guess it should refer to Section 4.4.
>
> Correct — the in-text reference "Figure 4.2" on page 8 should read "Figure 5." We will fix this.
>
> > I believe it poses a considerable limitation on the study. I recommend that the authors extend the proposed PGT framework to at least one more related benchmark.
>
> We have already included a cross-domain evaluation on OpenVLA/LIBERO-goal in Appendix A, covering 10 robotic manipulation tasks with a 7B OpenVLA model. PGT achieves 83.5% average success rate, matching GRAPE while requiring only 10 trajectory pairs per task. We will promote this result into the main text.
>
> ---
>
> We believe addressing these points — particularly the clarifications on Table 1, figure references, and the cross-domain results — will substantially strengthen the paper, and we hope the reviewer will consider raising the score.

---

> > ### Author Rebuttal · Reviewer_qDAX · 2026-04-01
> >
> > Author have kindly addressed my concerns and provided clarifications to their submission. I feel convinced and confident now to raise my score to 4/5.

---

> > > ### Author Response · Authors · 2026-04-02
> > >
> > > We sincerely thank the reviewer for the positive reassessment and for confirming that the concerns have been fully resolved. We truly appreciate the time and effort invested in reviewing our work — the feedback has meaningfully improved the paper.
> > >
> > > We noticed that the current score still reflects the original rating. Given the reviewer's kind indication of raising the score, we gently wanted to flag this in case the update has not yet been submitted. We completely understand if this is simply a matter of timing, and we are grateful for the reviewer's support either way.
> > >
> > > Thank you again for the constructive and encouraging review process.

---

### Decision · Program_Chairs · 2026-04-30

**Decision:**

Accept (regular)

**Comment:**

This paper proposes Preference Goal Tuning (PGT), a post-training framework that adapts frozen goal-conditioned policies by optimizing only the latent goal embedding with trajectory-level preference feedback. The overall review outcome is positive. Two reviewers were positive and became more favorable after rebuttal, while one reviewer remained somewhat skeptical despite acknowledging that several concerns were addressed.

Reviewers agreed that the paper studies an interesting and practical problem, and that the latent-control perspective is a meaningful way to separate task alignment from policy dynamics. Additionally, the comparison showing better OOD behavior than full fine-tuning and the authors demonstrated that the method can serve as a lightweight interface for sequential adaptation and planner-controller decomposition. Reviewers also viewed the method as sample-efficient.

The rebuttal usefully clarified missing presentation details, the role of reward-based preference proxies, the behavior-cloning baseline, PEFT comparisons, and the existence of a cross-domain evaluation. Concerns about missing baselines, similarity to soft-prompt-style adaptation, and insufficient comparison to PEFT methods were substantially addressed by the rebuttal. The main remaining weakness is that the central generalization claim is still supported primarily by Minecraft experiments, with the robotics result not presented as prominently as some reviewers expected.

Overall, I lean toward acceptance.